# Disruption of the TCA cycle reveals an ATF4-dependent integration of redox and amino acid metabolism

Dylan Gerard Ryan[1], Ming Yang[1], Hiran A Prag[2,3], Giovanny Rodriguez Blanco[4], Efterpi Nikitopoulou[1], Marc Segarra-Mondejar[1], Christopher A Powell[2], Tim Young[1], Nils Burger[2], Jan Lj Miljkovic[2], Michal Minczuk[2], Michael P Murphy[2], Alex von Kriegsheim[4], Christian Frezza[1]*†

[1]MRC Cancer Unit, University of Cambridge, Hutchison MRC Research Centre, Cambridge Biomedical Campus, Cambridge, United Kingdom; [2]MRC Mitochondrial Biology Unit, University of Cambridge, Cambridge Biomedical Campus, Cambridge, United Kingdom; [3]Department of Medicine, University of Cambridge, Cambridge Biomedical Campus, Cambridge, United Kingdom; [4]Edinburgh Cancer Research UK Centre, Institute of Genetics and Cancer, Edinburgh, United Kingdom

*For correspondence:
cf366@cam.ac.uk; christian.
frezza@uni-koeln.de

Present address: †CECAD
Research Center, Faculty of
Medicine, University Hospital
Cologne, Cologne, Germany

Competing interest: The authors declare that no competing interests exist.

**Summary** The Tricarboxylic Acid (TCA) Cycle is arguably the most critical metabolic cycle in physiology and exists as an essential interface coordinating cellular metabolism, bioenergetics, and redox homeostasis. Despite decades of research, a comprehensive investigation into the consequences of TCA cycle dysfunction remains elusive. Here, we targeted two TCA cycle enzymes, fumarate hydratase (FH) and succinate dehydrogenase (SDH), and combined metabolomics, transcriptomics, and proteomics analyses to fully appraise the consequences of TCA cycle inhibition (TCAi) in murine kidney epithelial cells. Our comparative approach shows that TCAi elicits a convergent rewiring of redox and amino acid metabolism dependent on the activation of ATF4 and the integrated stress response (ISR). Furthermore, we also uncover a divergent metabolic response, whereby acute FHi, but not SDHi, can maintain asparagine levels via reductive carboxylation and maintenance of cytosolic aspartate synthesis. Our work highlights an important interplay between the TCA cycle, redox biology, and amino acid homeostasis.

## Editor's evaluation

Ryan et al., compare the effects of acute FH or SDH inhibition with genetic ablation of FH or SDH in kidney epithelial cells. The consider how each intervention affects metabolite levels, alters the fate of labeled nutrients, and how it influences the transcriptome and proteome. This includes showing that disrupting the TCA cycle has a large effect on amino acid metabolism, and activates a stress response to maintain redox and amino acid homeostasis.

## Introduction

Most of the energy-rich adenosine triphosphate (ATP) generated in metabolism is provided by the complete oxidation of key fuel molecules to $CO_2$ in mitochondria. This catabolic process primarily occurs in the mitochondrial matrix via a series of reactions known as the Krebs cycle or the tricarboxylic acid (TCA) cycle (*Krebs and Johnson, 1980*; *Owen et al., 2002*; *Saraste, 1999*). In addition, the TCA cycle is a source of precursors for the synthesis of many other biological molecules, such as nonessential amino acids (NEAAs), lipids, nucleotide bases and porphyrin (*Owen et al., 2002*).

Therefore, any dysfunction of the TCA cycle is expected to elicit profound metabolic reprogramming of the cell beyond defective ATP generation. FH is a homotetrameric enzyme localized to the mitochondrial matrix and cytosol that catalyzes the stereospecific hydration of fumarate to L-malate (*Tuboi et al., 1986*). SDH, also known as complex II of the electron transport chain (ETC), is a heterotetrameric complex composed of four different subunits, SDHA, SDHB, SDHC, and SDHD (*Yankovskaya et al., 2003*). SDH localizes to the inner mitochondrial membrane (IMM) and catalyzes the oxidation of succinate to fumarate (*Schultz and Chan, 2001*; *Yankovskaya et al., 2003*). SDH is the only TCA cycle enzyme that is also a component of the ETC and so provides a physical link between the TCA cycle and oxidative phosphorylation (OXPHOS). Interestingly, mutations of both FH and SDH predispose kidney tubuluar epithelium to transformation (*Sciacovelli et al., 2020*), indicating that the kidney can be particularly affected by TCA cycle dysfunction.

Mitochondrial stress activates a retrograde signaling pathway to communicate metabolic dysfunction to the nucleus, a process first described in yeast over 20 years ago (*Butow and Avadhani, 2004*). However, only recently has mitochondrial stress signalling in mammals been linked to the integrated stress response (ISR) and the transcription factor Atf4 (*Bao et al., 2016*; *Mick et al., 2020*; *Pakos-Zebrucka et al., 2016*; *Quirós et al., 2017*). The ISR is an evolutionary conserved signaling network that responds to diverse cellular stresses, from amino acid deprivation to viral infection, and operates by reprogramming translation (general translation inhibitor via phosphorylation of the translational initiation factor eIF2α) and increasing the translation of specific stress-induced mRNAs, such as *Atf4* (*Pakos-Zebrucka et al., 2016*). While much is understood about the mechanisms of ISR signaling and Atf4 activation as a whole, we are only beginning to understand how mitochondria engage this pathway and how the ISR and Atf4 regulate the metabolic response to stress. Likewise, despite decades of research, a comprehensive investigation into the metabolic consequences and cellular response to TCA cycle dysfunction specifically in mammalian cells remains elusive. Here, we perform a comparative investigation into the consequences of acute TCAi in murine epithelial kidney cells. We identify a shared metabolic signature of TCAi, whereby enhanced glutathione synthesis is accompanied by a concomitant impairment in de novo proline and aspartate synthesis. Importantly, the selective disruption of mitochondrial thiol redox homeostasis is sufficient to recapitulate the convergent TCAi metabolic signature. Furthermore, acute TCAi phenocopied an amino acid deprivation-like state and activated the integrated stress response to counter redox and amino acid stress. Notably, acute SDHi led to a pronounced decrease in asparagine synthesis, an effect not observed with FHi, where reductive carboxylation and cytosolic aspartate synthesis maintain the asparagine pool. Finally, this work highlights a previously underappreciated role for oxidative TCA cycle activity and respiration in proline synthesis.

## Results

### TCA cycle inhibition promotes glutathione synthesis while impairing de novo proline and aspartate synthesis

To identify a common metabolic signature of TCA cycle dysfunction, we performed liquid chromatography-mass spectrometry (LC-MS)-based metabolomic analysis of kidney epithelial cells upon TCAi. To this aim, we assessed the response to acute TCAi with the FH inhibitor, FHIN-1, and the SDH inhibitors, Atpenin A5 (AA5) and Thenoyltrifluoroacetone (TTFA) (Acute model) (*Figure 1A*) as it enabled us to chart the early events of TCAi in a controlled fashion without adaptive compensations. We then compared it to cells in which *Fh1* or *Sdhb* have been genetically ablated (Chronic model). This analysis revealed substantial changes in the metabolome upon TCAi (*Figure 1—figure supplement 1A–1H*) with bona fide metabolic markers of FH and SDH deficiency observed (Figure S1E-H). For FHi, this included the accumulation of (S)–2-succinocysteine (2SC) and succinicGSH (*Figure 1—figure supplement 1E,G*), metabolites formed from fumarate-mediated alkylation of free cysteine and the major anti-oxidant GSH (*Zheng et al., 2015*). For SDHi, this included a significant accumulation in intracellular succinate levels (*Figure 1—figure supplement 1F,H*). Comparative analysis of all five conditions revealed that only two metabolites were significantly increased (*Figure 1B*), whilst eight metabolites were significantly decreased across all conditions (*Figure 1C*). The two significantly increased metabolites were GSH and oxidized GSH (GSSG), which suggested a perturbation in redox homeostasis when TCA cycle activity is inhibited. Of the eight decreased metabolites, notably, two

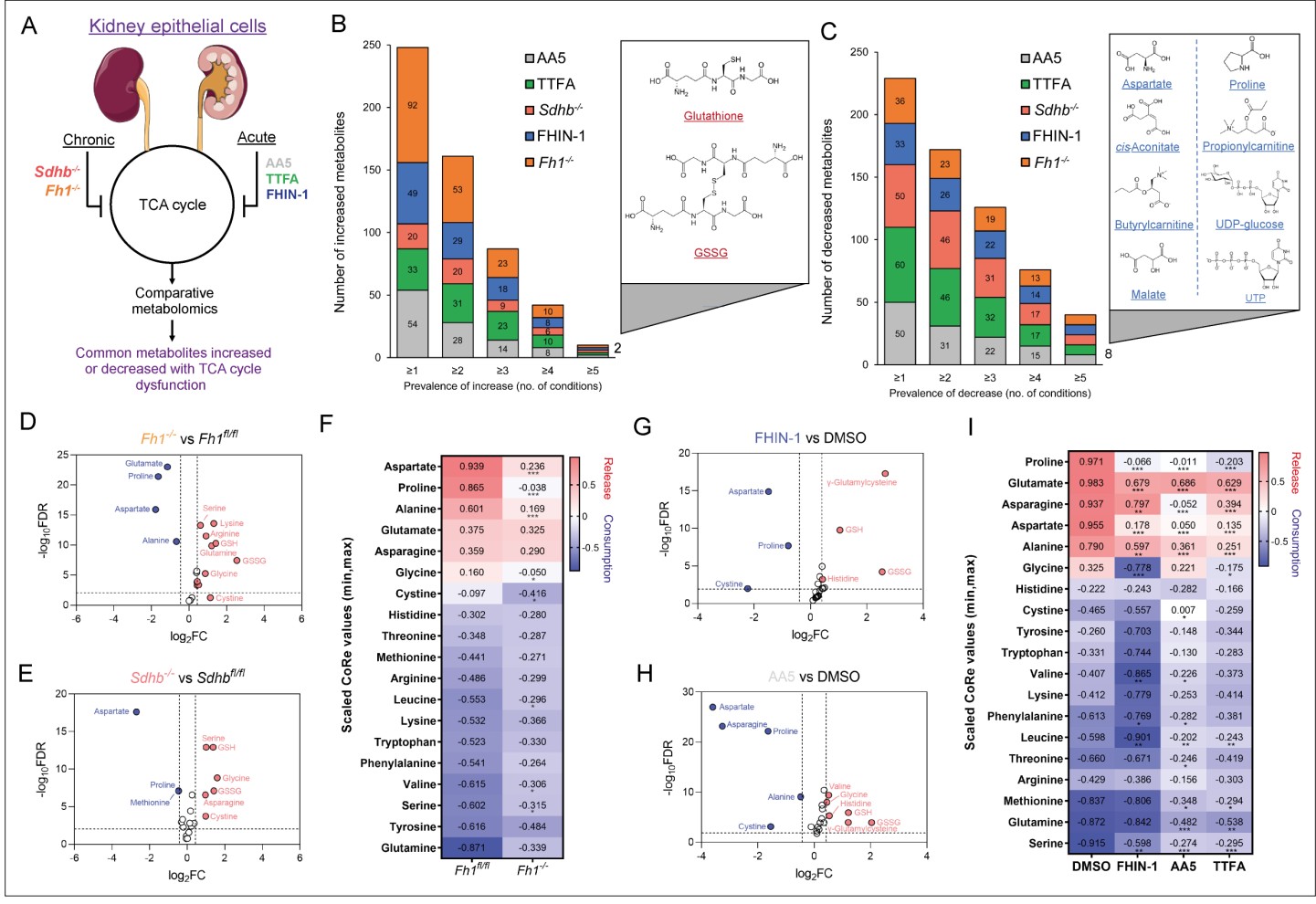

**Figure 1.** TCA cycle inhibition promotes cytosolic glutathione synthesis while impairing de novo proline and aspartate synthesis. (**A**) Schematic diagram of comparative metabolomic approach to assess the metabolic response of kidney epithelial cells to TCAi. (**B**) Significantly increased metabolites (Cut-off = 25% change in abundance, false discovery rate (FDR) = 5%) with genetic or pharmacological TCAi. (**C**) Significantly decreased metabolites. (**B–C**) (24 hr timepoint). (**D**) Volcano plot of glutathione-related metabolites and amino acids in *Fh1⁻/⁻* versus *Fh1ᶠˡ/ᶠˡ* cells. (**E**) Volcano plot of glutathione-related metabolites and amino acids in *Sdhb⁻/⁻* versus *Sdhbᶠˡ/ᶠˡ* cells. (**F**) Heatmap of mean scaled consumption-release (CoRe) intensity value of amino acids in *Fh1⁻/⁻* versus *Fh1ᶠˡ/ᶠˡ* cells. (**B–F**) (n = 5–10 independent biological replicates). (**G**) Volcano plot of glutathione-related metabolites and amino acids in FHIN-1 (20 µM)-treated *Fh1ᶠˡ/ᶠˡ* versus DMSO-vehicle control *Fh1ᶠˡ/ᶠˡ* cells. (**H**) Volcano plot of glutathione-related metabolites and amino acids in AA5 (1 µM)-treated *Fh1ᶠˡ/ᶠˡ* versus DMSO-treated *Fh1ᶠˡ/ᶠˡ* cells. (**G–H**) (24 hr timepoint) (n = 10–15 independent biological replicates). (**I**) Heatmap of mean scaled CoRe intensity value of amino acids in FHIN-1, AA5 and TTFA (500 µM) versus DMSO-treated *Fh1ᶠˡ/ᶠˡ* cells (24 hr timepoint) (n = 5 independent biological replicates). (**D–I**) p Value determined by multiple unpaired t-tests, corrected with two-stage step-up method of Benjamini, Krieger, and Yekutieli - FDR = 5%. p < 0.05*; p < 0.01**; p < 0.001***.

The online version of this article includes the following figure supplement(s) for figure 1:

**Figure supplement 1.** Metabolomic analysis of acute and chronic TCA cycle inhibition.

were TCA cycle metabolites (malate and *cis*-aconitate). Furthermore, the amino acids proline and aspartate were also decreased. Completing the list were propionylcarnitine and butyrylcarnitine, which are products of catabolic pathways known to feed the TCA cycle, while UTP and UDP-glucose are pyrimidine-associated metabolites (synthesized from aspartate) involved in glucose metabolism. Interestingly, while both acute FHi and SDHi impaired intracellular aspartate, proline and cystine levels (albeit to a different extent), only SDHi led to a significant decrease in intracellular asparagine levels (*Figure 1G–H* and *Figure 1—figure supplement 1I*). Of note, acute FHi or SDHi led to a decrease in aspartate and proline levels already after 1 hr of treatment (*Figure 1—figure supplement 1J,K*), which suggests that the impairment in amino acid synthesis precedes the changes in GSH metabolism. Finally, we observed that *Fh1⁻/⁻* and *Sdhb⁻/⁻* cells increased intracellular cystine, serine, glycine,

and asparagine levels, suggesting that chronic perturbations of the TCA cycle may induce adaptive compensatory metabolic changes (*Figure 1D–E*).

Considering the profound effect of TCAi on intracellular amino acid levels, we performed a consumption/release (CoRe) experiment upon acute FHi and SDHi. This analysis demonstrated a significant decrease in the release of aspartate, alanine, glutamate, and proline (*Figure 1I*) upon TCAi. Of note, while there was a small but significant decrease in asparagine release upon FHi, SDHi led to a far more pronounced decrease (*Figure 1I*). Coupled to the intracellular metabolomics, this result confirms a divergent metabolic response to the acute TCAi on asparagine synthesis depending on the enzyme targeted. Consistent with acute TCAi, *Fh1*[-/-] cells also demonstrated significant impairment in the release of aspartate, alanine and proline (*Figure 1F*). Overall, our metabolomic analyses identified perturbed redox homeostasis and amino acid metabolism, most notably a diminished capacity to synthesize aspartate and proline and an increase in GSH synthesis, as shared metabolic features of TCAi.

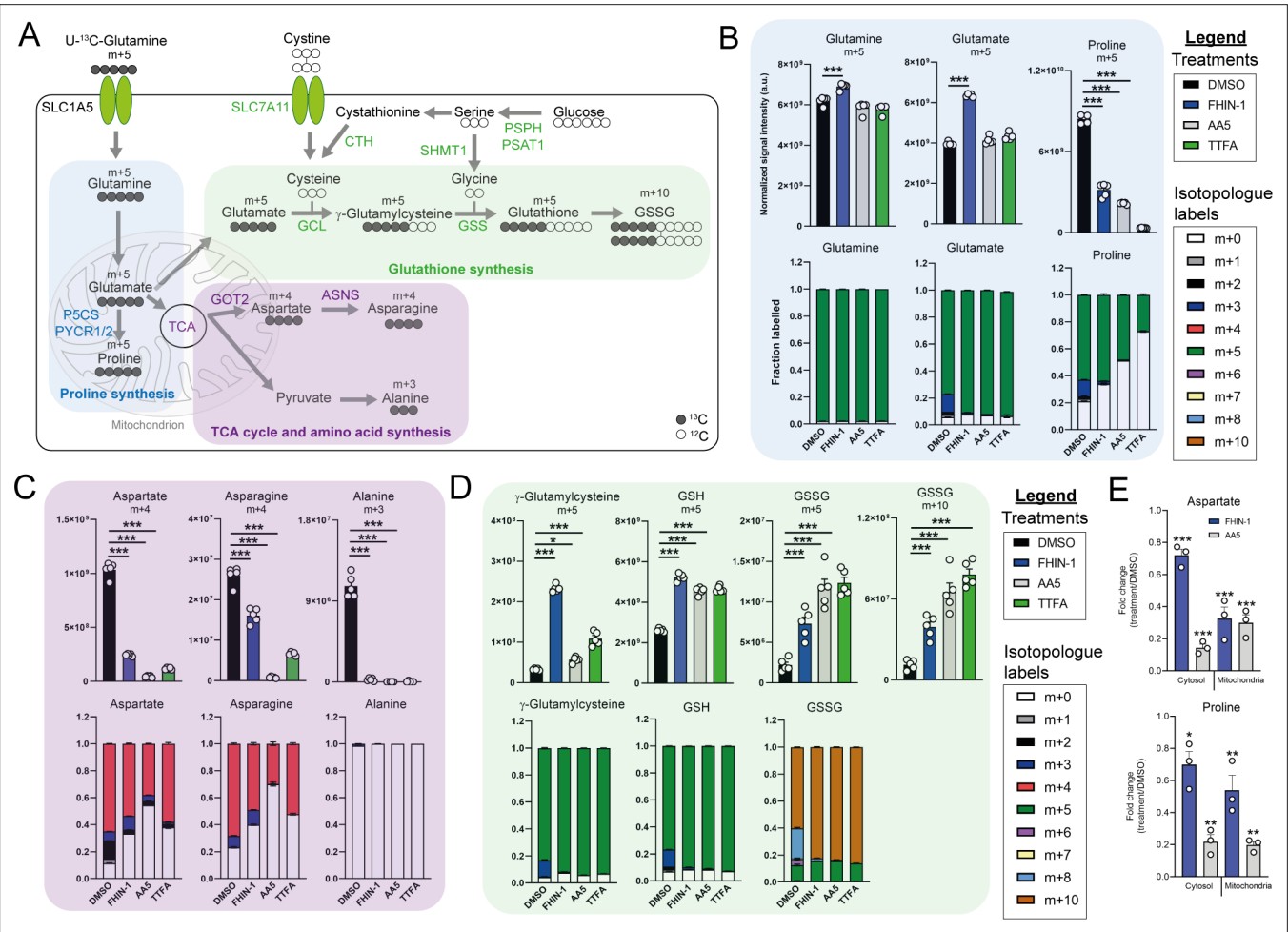

**Figure 2.** TCA cycle inhibition impairs glutamine-derived proline and aspartate synthesis but promotes cytosolic GSH synthesis. (**A**) Schematic diagram highlighting U-[13]C-glutamine tracing into distinct metabolic modules. (**B**) U-[13]C-glutamine tracing into proline (m + 5 labeling intensity and total isotopologue fraction distribution). (**C**) U-[13]C-glutamine tracing into aspartate, asparagine and alanine (m + 4 and m + 3 labeling intensity and total isotopologue fraction distribution). (**D**) U-[13]C-glutamine tracing into glutathione synthesis pathway (m + 5 and m + 10 labeling intensity and total isotopologue fraction distribution) (**B–D**) show DMSO, FHIN-1 (20 µM)-, AA5 (1 µM)- and TTFA (500 µM)-treated *Fh1*[fl/fl] cells (24 h timepoint) (n = 5 independent biological replicates). Data are mean ± standard error of mean (SEM). p Value determined by ordinary one-way ANOVA, corrected for multiple comparisons using Tukey statistical hypothesis testing. (**E**) Proline and aspartate level fold change in mitochondrial and cytosol fractions in FHIN-1 and AA5-treated versus DMSO-treated *Fh1*[fl/fl] cells (24 h timepoint) (n = 3 independent biological replicates). Data are mean ± standard error of mean (SEM). p Values determined by unpaired two-tailed t-test. p < 0.05*; p < 0.01**; p < 0.001***.

The online version of this article includes the following figure supplement(s) for figure 2:

**Figure supplement 1.** Glutamine fate tracing and compartment-specific metabolomics upon acute TCA cycle inhibition.

## TCA cycle inhibition and impaired mitochondrial thiol redox homeostasis rewire glutamine metabolism

To further characterize the metabolic reprogramming events associated with acute TCAi, we employed stable-assisted isotope tracing of U-$^{13}$C-glutamine with LC-MS analysis of our kidney epithelial cells, and focused on the three metabolic modules highlighted above, ie proline, aspartate, and GSH metabolism (*Figure 2A*).

Glutamine can be converted into glutamate by glutaminase and can then be diverted into several metabolic pathways, including GSH synthesis, proline synthesis, and the TCA cycle (*Figure 2A*). Stable-assisted isotope tracing revealed that a significant proportion of glutamate and proline are derived from glutamine as their pools consist primarily of the m + 5 isotopologue (*Figure 2B*). Total m + 5 labeling from glutamine significantly decreased in proline, suggesting an impairment in glutamate-derived proline synthesis upon TCAi (*Figure 2B*). Notably, m + 3 labeled proline, which is derived from α-ketoglutarate (α-KG) that has undergone one round of oxidative TCA cycle catabolism (*Figure 2—figure supplement 1A*), disappears with TCAi. U-$^{13}$C-glutamine tracing also revealed robust labeling of TCA cycle intermediates (*Figure 2—figure supplement 1B*) and an impairment in oxidative TCA cycling with FHi and SDHi, as determined by a decrease in m + 4 and m + 2-labeled citrate and *cis*-aconitate, m + 3-labeled α-KG and 2-hydroxyglutarate (2-HG) and m + 2-labeled succinate, malate, fumarate, and aspartate (*Figure 2—figure supplement 1B*). Aspartate is directly synthesized from TCA cycle-derived oxaloacetate via glutamate-aspartate transaminase 2 (GOT2), downstream of FH and SDH in mitochondria (*Birsoy et al., 2015*; *Cardaci et al., 2015*; *Sullivan et al., 2015*). Aspartate is subsequently exported to the cytosol where it can also be converted into asparagine via the action of asparagine synthetase (ASNS) (*Zhang et al., 2014*). We observed a significant decrease in m + 4-labeled aspartate and asparagine upon TCAi, suggesting an impairment in mitochondrial-synthesized aspartate (*Figure 2C*). It is worth noting that SDHi was more effective at decreasing m + 4-labeled aspartate than FHi, explaining the lower levels of asparagine synthesis observed with SDHi. Acute FHi promoted reductive carboxylation, as determined by an increase in m + 5-labeled isotopologue fractional enrichment in α-KG, 2-HG, citrate and *cis*-aconitate and in m + 3-labeled malate, aspartate and asparagine (which arises from the cytosolic synthesis of aspartate via GOT1) (*Birsoy et al., 2015*; *Figure 2C* and *Figure 2—figure supplement 1B*). Despite a reduction in the m + 3 labeling, the total pools of malate, aspartate, and asparagine were maintained (*Figure 2—figure supplement 1C*). In contrast to FHi, SDHi significantly decreased reductive carboxylation and fully impaired cytosolic aspartate and asparagine synthesis (*Figure 2—figure supplement 1C*). A complete loss of m + 3 labeled alanine was also observed with TCAi, which may be derived from oxaloacetate-derived phosphoenolpyruvate (PEP) or from m + 3-labeled malate as a consequence of reductive carboxylation (*Figure 2—figure supplement 1A*). Consistent with elevated levels of cytosolic GSH synthesis following TCAi, an increase in the intensity of m + 5 labeling in γ-glutamylcysteine, GSH and GSSG and m + 10-labeled GSSG was observed (*Figure 2D*).

This tracing analysis was further supported by combining metabolomics of mitochondrial and cytosolic fractions. Indeed, whilst mitochondrial aspartate levels where equally reduced by both FHi and SDHi, SDHi led to a more pronounced decrease in the cytosolic fraction of aspartate (*Figure 2E*). Additionally, asparagine levels were maintained in the cytosol with FHi, but decreased with SDHi (*Figure 2—figure supplement 1D*). Therefore, asparagine levels are maintained upon FHi in part from a less severe depletion of cytosolic aspartate levels and from the maintenance of reductive carboxylation and GOT1-derived aspartate synthesis. Fractionation also revealed a decrease in proline upon TCAi, and an increase in GSSG levels with SDHi and in succinicGSH with FHi across both fractions (*Figure 2E* and *Figure 2—figure supplement 1D*).

Due to increased oxidative stress with TCAi, as determined by a significant increase in GSSG levels when compared to GSH levels (*Figure 3—figure supplement 1A*) and the aforementioned accumulation of succinicGSH adducts with FHi, we investigated whether cytosolic GSH synthesis and amino acid metabolism were sensitive to perturbations in mitochondrial thiol redox homeostasis. To achieve this, we utilized a recently developed tool, termed mitoCDNB, to selectively deplete mitochondrial GSH pools and inhibit thioredoxin reductase 2 and peroxiredoxin 3, thus impairing mitochondrial thiol redox homeostasis (*Booty et al., 2019*; *Cvetko et al., 2021*). We confirmed the previously-described formation of the mitoCDNB-GSH adduct (MitoGSDNB) upon treatment with mitoCDNB (*Booty et al., 2019*), thus validating the approach (*Figure 3A*). Similar to TCAi, impaired mitochondrial thiol

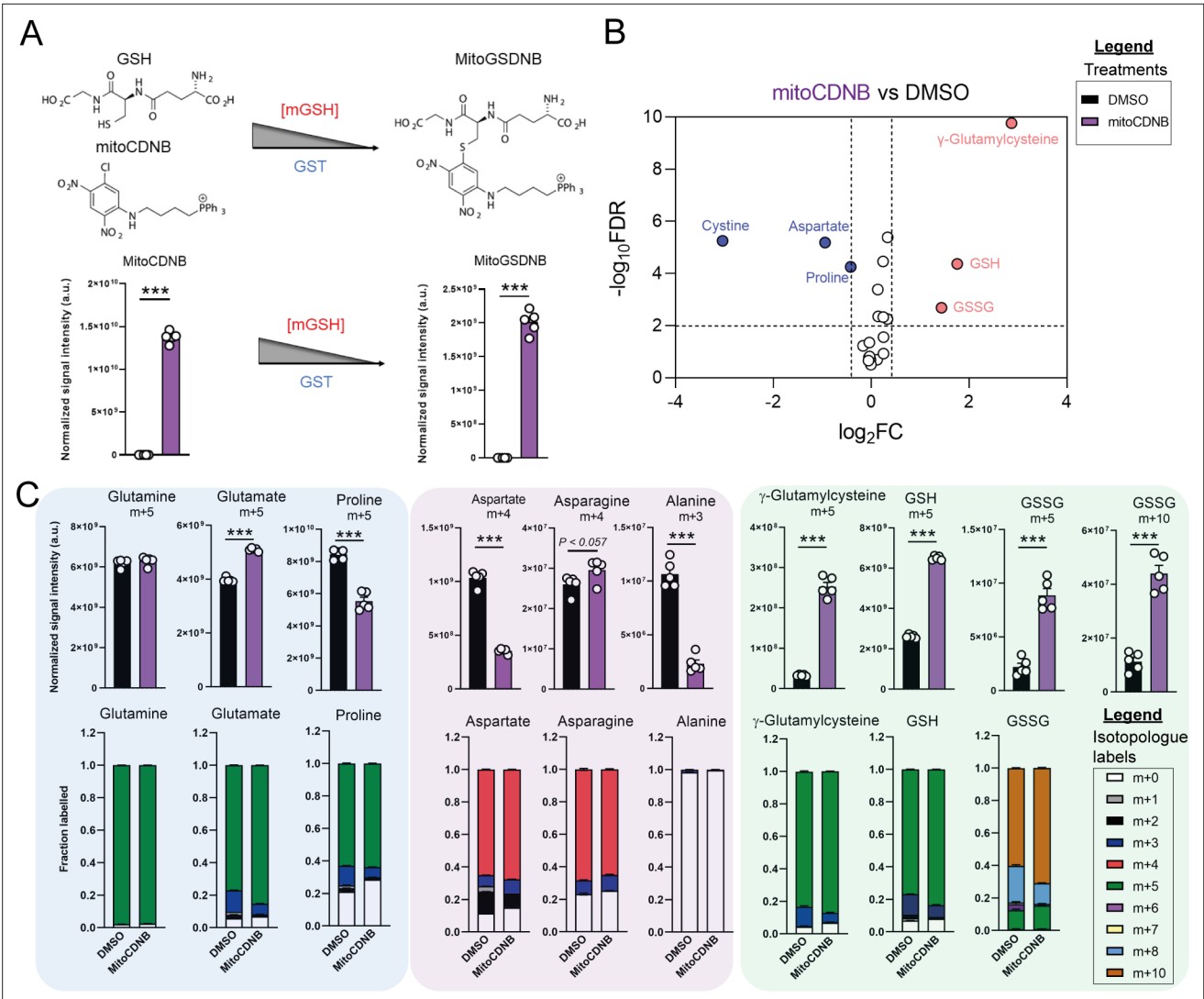

**Figure 3.** Disrupted mitochondrial thiol redox homeostasis phenocopies TCA cycle inhibition. (**A**) Schematic diagram highlighting mito 1-Chloro-2,4-dinitrobenzene (mitoCDNB) reaction with glutathione and the formation of mito 1-S-glutathionyl-2,4-dinitrobenzene (MitoGSDNB) and intensity levels in DMSO-treated and mitoCDNB (10 μM)-treated *Fh1fl/fl* cells. (**B**) Volcano plot of glutathione-related metabolites and amino acids in mitoCDNB-treated *Fh1fl/fl* versus DMSO-treated *Fh1fl/fl* cells. p Value determined by multiple unpaired t-tests, corrected with two-stage step-up method of Benjamini, Krieger and Yekutieli - FDR = 5%. (**C**) U-$^{13}$C-glutamine tracing (m + 10, m + 5, m + 4 and m + 3 labeling intensity and total isotopologue fraction distribution) in DMSO and mitoCDNB-treated *Fh1fl/fl* cells. (**A–C**) (24 hr timepoint) (n = 5–10 independent biological replicates). (**A and C**) Data are mean ± SEM. p Value determined by unpaired two tailed t-test or ordinary one-way ANOVA corrected for multiple comparisons using Tukey statistical hypothesis testing. p < 0.05*; p < 0.01**; p < 0.001***.

The online version of this article includes the following figure supplement(s) for figure 3:

**Figure supplement 1.** Glutamine fate tracing upon mitochondrial GSH depletion.

redox homeostasis significantly decreased proline and aspartate levels and increased GSH synthesis (*Figure 3B*). Furthermore, mitoCDNB decreased m + 5 labeling in proline and m + 4 labeling in aspartate while increasing m + 5 labeling inγ-glutamylcysteine, GSH and GSSG and m + 10-labeled GSSG (*Figure 3C*) upon incubation with U-$^{13}$C-glutamine. The decrease in aspartate with mitoCDNB is likely due to reduced oxidative TCA cycling (*Figure 3C* and *Figure 3—figure supplement 1B*). Interestingly, m + 4 labeled asparagine was unaffected with mitoCDNB and similarly to FHi, we observe an increase in reductive carboxylation, cytosolic aspartate synthesis, and an increase in m + 3-labeled asparagine despite lower levels of m + 3-labeled aspartate (*Figure 3—figure supplement 1C*). Overall, this

data supports the idea that impaired mitochondrial redox homeostasis may causally contribute to the changes in amino acid metabolism observed upon TCAi.

## Impaired respiration underlies defect in proline biosynthesis

We then investigated the metabolic determinants of the unexpected defects in proline metabolism elicited by TCAi. Given that glutamate is an essential source for proline biosynthesis, TCA cycle anaplerosis, and GSH synthesis, we hypothesized that the defect in proline might arise from differences in glutamate apportioning into these pathways (*Figure 2A*). We therefore tested whether the γ-glutamylcysteine ligase (GCL) inhibitor L-buthionine-sulfoximine (BSO), which blunted GSH synthesis (*Figure 4A*), would restore proline levels. In support of the glutamate apportioning hypothesis, BSO treatment of kidney epithelial cells increased glutamate, aspartate and asparagine abundance, whilst a trend in increased proline was also observed (*Figure 4B*). In contrast, when TCAi was coupled with impaired GSH synthesis, it led to significantly higher intracellular glutamate and cystine levels but failed to rescue the decrease in proline (*Figure 4C and D*). In fact, proline levels decreased further, suggesting that glutathione synthesis is required to support proline synthesis when TCA cycle activity is interrupted and redox stress increases. Likewise, alanine, asparagine, and aspartate levels were also significantly lower with a combination of FHi and BSO (*Figure 4C*), whilst SDHi and BSO lead to a further drop in asparagine (*Figure 4D*). Similar to BSO treatment, the incubation with ethylGSH (eGSH), which supplements the GSH pool and spares glutathione-related metabolites, such as γ-glutamylcysteine, cystine, and cys-gly (*Figure 4—figure supplement 1A*), increased intracellular glutamate levels when combined with TCAi but failed to rescue the decrease in proline (*Figure 4—figure supplement 1B*). Together, these findings show that intracellular glutamate apportioning does not explain the defect in de novo proline synthesis with TCAi.

To further understand what could give rise to the observed defects in proline synthesis with TCAi, we decided to investigate changes in both the redox and bioenergetic state of the cell. To achieve this, we measured the NADH/NAD$^+$, NADPH/NADP$^+$, and ATP/ADP ratio with TCAi on its own or coupled to BSO or eGSH treatment (*Figure 4E*). Intriguingly, we observed a significant increase in the NADH/NAD$^+$ ratio with both FHi (~12 fold) and SDHi (~4.65 fold), but the increase was more than double with FHi (*Figure 4E*). This difference was also reflected by the elevated lactate/pyruvate ratio observed with FHi compared to SDHi, whereas the decrease in the citrate/pyruvate ratio was similar between both conditions (*Figure 4—figure supplement 1D*). Furthermore, the NADH/NAD$^+$ ratio was decreased with BSO co-treatment suggesting an essential interplay between GSH synthesis and the redox state of the cell upon TCAi (*Figure 4E*). TCAi also led to a significant decrease in the NADPH/NADP$^+$ ratio (*Figure 4E*). The NADPH/NADP$^+$ ratio is a readout of oxidative stress given the role of NADPH in supporting anti-oxidant defence systems and in the reduction of cystine to cysteine for GSH synthesis (*Zheng et al., 2015*). Indeed, we observed an increase in the NADPH/NADP$^+$ ratio (and an increase in cystine) with TCAi and BSO co-treatment, while eGSH supplementation prevented a decrease in the NADPH/NADP$^+$ ratio with TCAi (*Figure 4E*). This data suggests that TCAi places cells under conditions of oxidative stress and that the drop in NADPH is due in part to an increased requirement of the cell to convert oxidized cystine to reduced cysteine to support GSH synthesis. Recently, NADPH has been implicated as the major cofactor supporting mitochondrial proline biosynthesis and this could partly explain the decrease in proline (*Tran et al., 2021*; *Zhu et al., 2021*). However, neither BSO or eGSH could rescue proline synthesis despite increasing intracellular glutamate or re-establishing NADPH levels (*Figure 4B and C*). Whilst not precluding the NADPH/NADP$^+$ ratio as being an important mitochondrial factor governing this response, it suggested other mechanisms could also be at play with TCAi.

The main function of the TCA cycle is to support NADH generation for OXPHOS. As expected, TCAi led to a significant decrease in the ATP/ADP ratio (*Figure 4E*) and a decrease in mitochondrial respiration and ATP synthesis, as assessed with a Seahorse flux analyzer (*Figure 4F*). Unlike the NADPH/NADP$^+$ ratio, BSO and eGSH treatments failed to rescue the decrease in the ATP/ADP ratio, and FHi and BSO treatment actually led to a further decrease (*Figure 4E*). Given the importance of ATP as a cofactor for the proline synthetic enzyme Aldh18a1 (also known as P5C synthetase), this data suggested that impaired respiration and ATP availability was a factor governing reduced proline biosynthesis with TCAi. To validate this hypothesis, we treated cells with the $F_1F_0$-ATP synthase inhibitor oligomycin. In line with our hypothesis, we observed a significant decrease in the ATP/ADP ratio

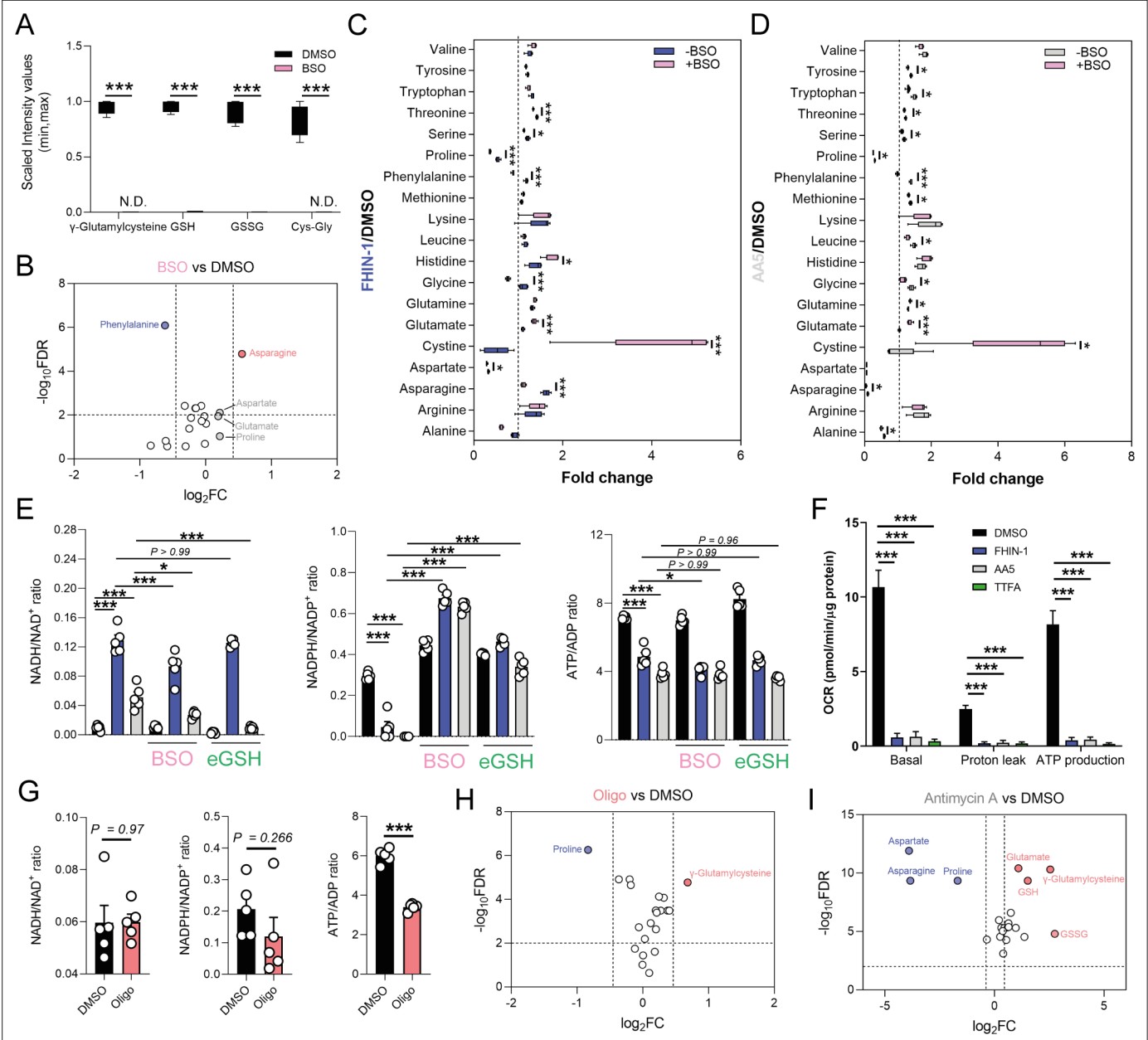

**Figure 4.** Impaired mitochondrial respiration underlies defect in de novo proline synthesis. (**A**) Scaled intensity values of glutathione-related metabolites in BSO (500 µM)-treated and DMSO-treated *Fh1fl/fl* cells (24 hr timepoint). (**B**) Volcano plot of amino acids in BSO-treated versus DMSO-treated *Fh1fl/fl* cells (24 hr timepoint). (**C**) Interleaved box and whiskers plot of amino acids with FHIN-1-treated or FHIN-1 and BSO co-treatment of *Fh1fl/fl* cells (24 hr timepoint). (**D**) Interleaved box and whiskers plot of amino acids with AA5-treated or AA5 and BSO co-treatment of *Fh1fl/fl* cells. (**E**) NADH/NAD+, NADPH/NADP+ and ATP/ADP ratio in DMSO, FHIN-1- and AA5-treated *Fh1fl/fl* cells ± BSO or ethylGSH co-treatment. (**F**) OCR assessed by Seahorse analyser in DMSO, FHIN-1-, AA5-, and TTFA-treated *Fh1fl/fl* cells (**A–F**) (n = 5 independent biological replicates). (**E–F**) Data are mean ± SEM. p Value determined by ordinary one-way ANOVA, corrected for multiple comparisons using Tukey statistical hypothesis testing. p < 0.05*; p < 0.01**; p < 0.001***. (**G**) NADH/NAD+, NADPH/NADP+, and ATP/ADP ratio in DMSO and Oligomycin (10 µM)-treated *Fh1fl/fl* cells (1 hr timepoint). Data are mean ± SEM. p Values determined by unpaired two-tailed t-test. p < 0.05*; p < 0.01**; p < 0.001***. (**H**) Volcano plot of glutathione-related metabolites and amino acids in Oligomycin-treated versus DMSO-treated *Fh1fl/fl* cells (1 h timepoint) (**G–H**) (n = 5 independent biological replicates). (**I**) Volcano plot of glutathione-related metabolites and amino acids in Antimycin A (2 µM)-treated versus DMSO-treated *Fh1fl/fl* cells (24 hr timepoint) (n = 5 independent biological replicates). (A-D; **H–I**) p Value determined by multiple unpaired t-tests, corrected with two-stage step-up method of Benjamini, Krieger, and Yekutieli - FDR = 5%.

The online version of this article includes the following figure supplement(s) for figure 4:

**Figure supplement 1.** eGSH supplementation upon acute TCA cycle inhibition.

(*Figure 4G*) and proline (*Figure 4H*). We also observed an increase in γ-glutamylcysteine (*Figure 4H*) and no significant change in the NADH/NAD$^+$ or NADPH/NADP$^+$ ratio (*Figure 4G*), although a trend toward a decrease in the NADPH/NADP$^+$ ratio was observed. Given the requirement of ATP for NADK2-mediated phosphorylation of NAD$^+$ to generate NADP$^+$ in mitochondria (*Tran et al., 2021*; *Zhu et al., 2021*), it's likely that a combined reduction of mitochondrial ATP and NAD(P)H give rise to the defect in proline synthesis. Further support for mitochondrial respiration involvement in regulating de novo proline synthesis came from the inhibition of complex III with antimycin A, which significantly decreased aspartate, asparagine and proline, while promoting GSH synthesis (*Figure 4I*). mitoCDNB had no significant effect on the overall ATP/ADP ratio, but increased the lactate/ pyruvate and the NADH/NAD$^+$ ratio (*Figure 3—figure supplement 1D*), indicating a minor defect in mitochondrial complex I and respiration with no alterations in the ATP/ADP ratio. This result is also consistent with increased asparagine levels (*Figure 3—figure supplement 1C*) and lower levels of proline inhibition

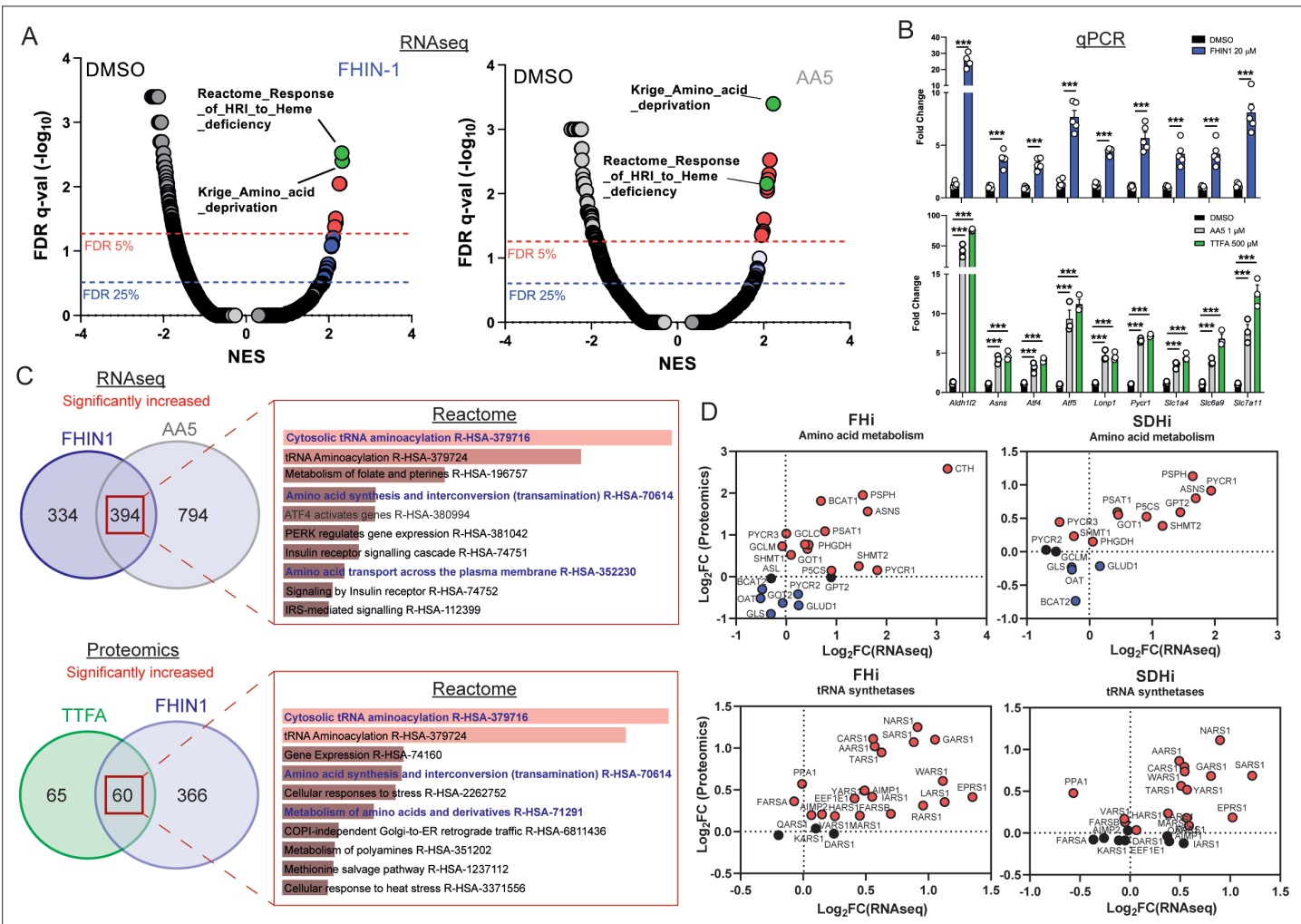

**Figure 5.** TCA cycle inhibition mimics an amino acid deprivation response and promotes compensatory reprogramming. (**A**) GSEA of RNAseq data from FHIN-1- and AA5-treated versus DMSO-treated *Fh1$^{fl/fl}$* cells. (**B**) Quantitative PCR results from FHIN-1-, AA5-, TTFA-, and DMSO-treated *Fh1$^{fl/fl}$* cells (24 hr timepoint) (n = 3–5 independent biological replicates). Data are mean ± SEM. p Value determined by unpaired two-tailed t-test or ordinary one-way ANOVA, corrected for multiple comparisons using Tukey statistical hypothesis testing. p < 0.05*; p < 0.01**; p < 0.001***. (**C**) ORA of overlapping transcripts significantly increased in FHIN-1 and AA5-treated *Fh1$^{fl/fl}$* cells and between overlapping proteins significantly increased in FHIN-1 and TTFA-treated *Fh1$^{fl/fl}$* cells (24 h timepoint) (n = 3–5 independent biological replicates). Reactome pathways ranked by p value. (**D**) Comparison of amino acid metabolism and tRNA synthetase transcript and protein levels.

The online version of this article includes the following figure supplement(s) for figure 5:

**Figure supplement 1.** Over representation analysis of significantly decreased targets upon acute TCA cycle inhibition.

with mitoCDNB (*Figure 3B*). Overall, TCAi uncovered an underappreciated interconnection between GSH and proline metabolism, underpinned by both metabolic and bioenergetics cues.

## TCA cycle inhibition mimics an amino acid deprivation-like response and promotes compensatory reprogramming

We then intended to investigate whether any transcriptional or translational response contributed to the metabolic changes we described upon TCAi. To achieve this, we performed mRNA-seq on FHIN-1- and AA5-treated cells (*Figure 5A and C* and *Figure 5—figure supplement 1A*) and proteomics analysis on FHIN-1- and TTFA-treated cells (*Figure 5C* and *Figure 5—figure supplement 1B,C*). Both treatments resulted in highly significant changes in the transcriptome and gene set enrichment analysis (GSEA) (*Subramanian et al., 2005*) identified robust activation of an amino acid deprivation response, in line with the decrease in aspartate and proline we observed. We also revealed a heme regulated inhibitor (HRI) stress response signature upon TCAi (*Figure 5A*). We validated several of the most significantly increased targets via qPCR (*Figure 5B*) and confirmed that TTFA similarly increased the targets when compared with AA5 (*Figure 5B*). To identify a common transcriptional and proteomic signature of TCAi, we compared the significantly increased and decreased genes/proteins and constructed comparative Venn diagrams (*Figure 5C* and *Figure 5—figure supplement 1A,B*). Overrepresentation analysis (ORA) of the significantly increased genes/proteins using Enrichr (*Chen et al., 2013*) revealed that cytosolic tRNA aminoacylation and amino acid metabolism were among the most enriched pathways using the Reactome database (*Figure 5C*) again lending support to the idea that perturbed amino acid metabolism was a key outcome of interrupted TCA cycle activity. We also observed a suppression of cholesterol biosynthesis (*Figure 5—figure supplement 1A*) consistent with previous reports using respiration inhibitors (*Mick et al., 2020*), while mitoribosome subunits and respiration-associated proteins were found to decrease in abundance with TCAi (*Figure 5—figure supplement 1B*). However, despite this apparent decrease in mitoribosome subunits, we failed to observe a decrease in 35S-methionine labeling of mitochondrial proteins with TCAi (*Figure 5—figure supplement 1D*), which suggests that mitochondrial protein synthesis is resistant to both impaired respiration and a decrease in the components of the mitochondrial translation apparatus over the tested timeframe (24 h). Notably, FHi led to a more pronounced decrease in Complex I biogenesis and subunits (*Figure 5—figure supplement 1C*) consistent with the previous reports on FH-deficiency regulating iron-sulphur cluster biogenesis and complex I activity (*Tong et al., 2011*; *Tyrakis et al., 2017*), while also explaining the noticeable differences in the NADH/NAD$^+$ ratio between FHi and SDHi (*Figure 4E*).

Comparing the log$_2$ fold change of enzymes involved in amino acid metabolism detected in both the proteomics and RNAseq highlighted an increase in the transcript and protein level of Asns with both FHi and SDHi (*Figure 5D*). Likewise, GOT1 levels were also increased (*Figure 5D*). This data suggested that cytosolic asparagine synthesis was maintained with FHi in part via reductive carboxylation and GOT1-directed aspartate synthesis, but also due to increased Asns expression. It also emphasized the importance of cytosolic aspartate synthesis, as acute SDHi robustly increased Asns but failed to maintain asparagine synthesis. A clear upregulation of PSPH, PSAT1, PHGDH, and SHMT1/2 is also observed with TCAi (*Figure 5D*), which may support GSH synthesis via the provision of glycine (*Figure 2A*). Interestingly, TCAi led to a robust increase in the cytosolic proline synthetic enzyme, Pycr3, at the protein level. Consistent with a more pronounced decrease in proline levels with SDHi (*Figure 1G and H*, S1I), Aldh18a1 and Pycr1 were also upregulated to a greater extent at the protein level (*Figure 5D*). Furthermore, the most highly increased tRNA synthetase at the protein level with TCAi was the asparaginyl-tRNA synthetase (NARS1), while one of the most highly increased on the transcript level was the glutamate-proline tRNA synthetase (EPRS1). These data support the notion that a key metabolic module affected by TCA cycle dysfunction is amino acid metabolism, with a particular focus on proline and aspartate/asparagine. Finally, a notable increase in the cysteine (CARS)-, glycine (GARS)-, alanine (AARS)-, and serine(SARS)-tRNA synthetases along with enzymes associated with cysteine, glycine, serine, and alanine biosynthesis (CTH, SHMT1/2, PSAT1, PSPH1 and GPT2) with either FHi or SDHi (or both) (*Figure 5D*) lend further support to the idea TCAi promotes compensatory transcriptional/translational rewiring to support GSH and amino acid metabolism in order to counteract amino acid stress.

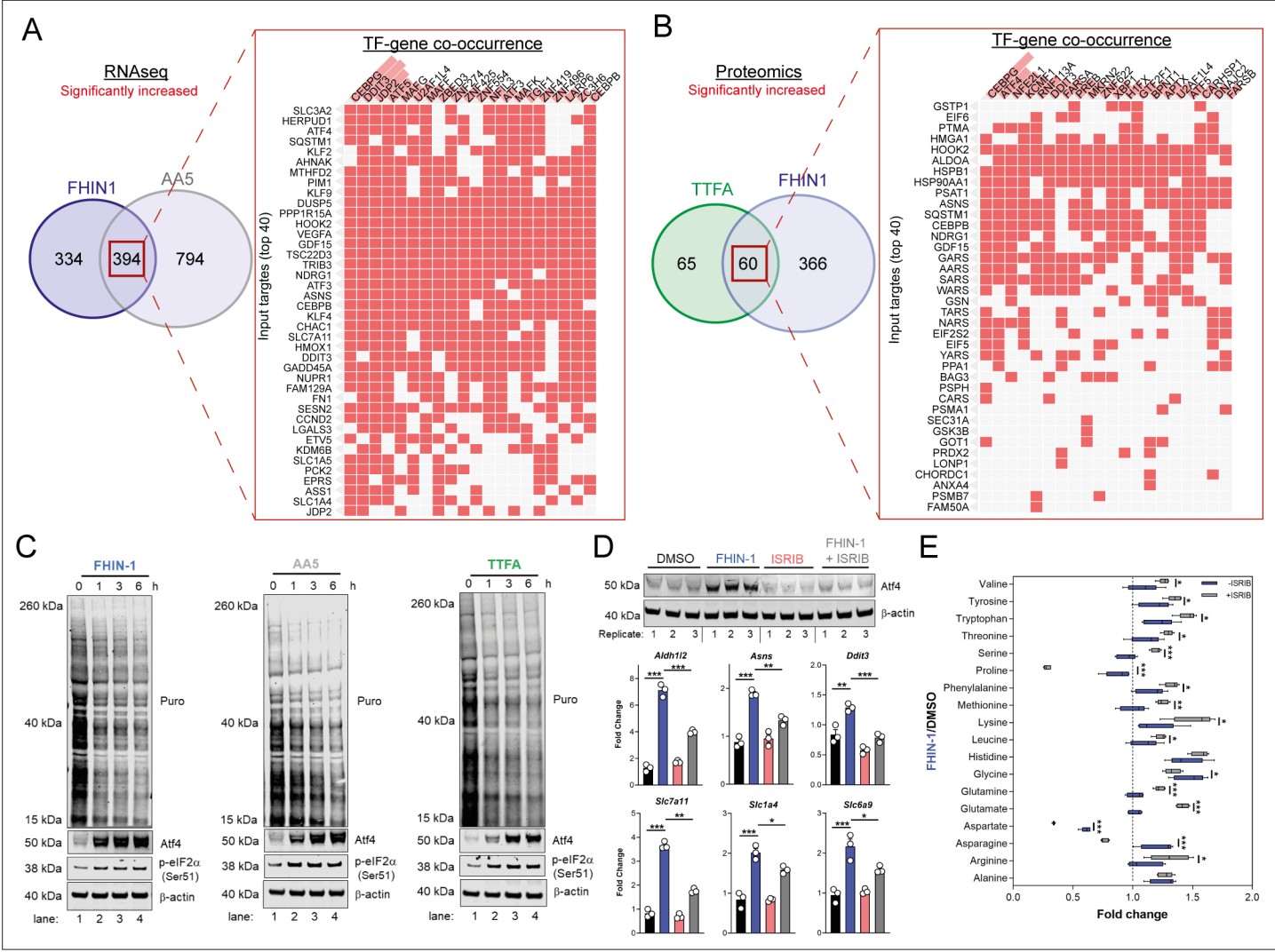

**Figure 6.** TCA inhibition activates the integrated stress response to counter amino acid and redox stress. (**A**) Clustergram showing Transcription-factor (TF)-gene co-occurrence ORA of overlapping transcripts significantly increased in FHIN-1 and AA5-treated *Fh1^fl/fl* cells. (**B**) Clustergram showing TF-gene co-occurrence ORA of overlapping transcripts significantly increased in FHIN-1 and TTFA-treated *Fh1^fl/fl* cells. (**A–B**) (24 hr timepoint) (n = 3–5 independent biological replicates). TFs are ranked by combined score (p value and Z score). (**C**) Western blot analysis of puromycin incorporation, Atf4, phopho-eIF2α and β-actin as loading control in FHIN-1, AA5 and TTFA timecourse treatment of *Fh1^fl/fl* cells (representative image of three independent biological replicates). (**D**) Western blot analysis of Atf4 and β-actin as loading control (6 hr timepoint) and qPCR analysis of Atf4-target genes (24 hr timepoint) (n = 3 independent biological replicates) in DMSO-, ISRIB (500 nM)-, FHIN-1 (10 μM), and FHIN-1+ ISRIB treated *Fh1^fl/fl* cells. Data are mean ± SEM. p value determined by ordinary one-way ANOVA, corrected for multiple comparisons using Tukey statistical hypothesis testing. (**E**) Interleaved box and whiskers plot of amino acids with FHIN-1-treated or FHIN-1- and ISRIB- co-treated *Fh1^fl/fl* cells (24 hr timepoint) (n = 5 independent biological replicates). p Value determined by multiple unpaired t-tests, corrected with two-stage step-up method of Benjamini, Krieger, and Yekutieli - FDR = 5%. p < 0.05*; p < 0.01**; p < 0.001***.

The online version of this article includes the following figure supplement(s) for figure 6:

**Figure supplement 1.** Atf4 activation upon acute TCA cycle inhibition and mitochondrial GSH depletion.

## TCA cycle inhibition activates Atf4 and the integrated stress response

To understand how the transcriptional changes induced upon TCAi were communicated to the nucleus, we performed transcription factor (TF) enrichment analysis and uncovered activation of several metabolic stress response TFs associated with the ISR, namely Ddit3/Chop, Atf5, Atf4, Atf3, Cebpb, and Cebpg (*Figure 6A and B*). Using a mitochondrial ISR/Atf4 gene signature curated from the literature (*Bao et al., 2016*; *Guo et al., 2020*; *Quirós et al., 2017*), we confirmed a significant enrichment of Atf4 targets with both FHi (*Figure 6—figure supplement 1A*) and SDHi (*Figure 6—figure*

*supplement 1B*) when compared to other hallmark genesets from the molecular signatures database (MsigDB) (*Subramanian et al., 2005*). To validate engagement of the ISR with TCAi, we probed for puromycinylated proteins and phosphorylation of eIF2α on serine 51, a readout of global translation rates and ISR activation (*Rabouw et al., 2019*), and demonstrated a time-dependent decrease in translation and increase in p-eIF2α as early as 1 hr post-treatment (*Figure 6C*). Furthermore, a robust and time-dependent increase in Atf4 stabilization was also observed (*Figure 6C*). In addition, mitochondrial GSH depletion with mitoCDNB also activated Atf4 to a similar extent as TCAi (*Figure 6—figure supplement 1C*).

To confirm a role for the ISR in regulating Atf4 stabilization and the expression of Atf4- targets with TCAi, we co-treated cells with FHIN-1 and the specific integrated stress response inhibitor (ISRIB) (*Sidrauski et al., 2015*), a potent antagonist of p-eIF2α. ISRIB works within a defined window of activation, inhibiting low level ISR activity but not affecting strong ISR signaling (*Rabouw et al., 2019*). Therefore, we used a lower dose of FHIN-1 (10 µM) and observed an attenuation in FHi-induced Atf4 stability and Atf4-target gene expression (*Figure 6D*). Importantly, this regulation was only strongly observed with FHi-induced Atf4 and not basal Atf4 levels (*Figure 6D*). This result confirmed that the ISR was involved in Atf4 stabilization and FHi-induced gene expression. In order to determine whether induction of the ISR regulated the metabolic response to TCAi, we treated wild-type cells with FHIN-1 or FHIN-1 and ISRIB (*Figure 6E*). While ISRIB treatment led to a significant decrease in proline (~25%) (*Figure 6—figure supplement 1D*) likely due to basal p-eIF2α (*Figure 6C*). Co-treatment of FHIN-1 and ISRIB led to a substantial decrease in intracellular proline (~75%) when compared to FHIN-1 alone (*Figure 6E*). Similarly, aspartate and asparagine levels were also decreased with co-treatment of FHIN-1 and ISRIB (*Figure 6E*) supporting the idea that the ISR was activated to compensate for the defect in the de novo synthesis of these amino acids. In addition to countering amino acid stress, we observed an increase in oxidative stress with FHi when the ISR was attenuated, as assessed by a significant increase in GSSG/GSH ratio (*Figure 6—figure supplement 1E*). In summary, TCAi triggers the ISR to counter amino acid and redox stress in kidney epithelial cells.

To evaluate the role of Atf4 in the response to TCAi, we ablated Atf4 expression using two independent siRNAs and confirmed a decrease at the protein and transcript level of Atf4 and Atf4-target genes (*Figure 7A*). Atf4 was stabilized to a certain extent basally in our cells and its ablation revealed that Atf4 is a key regulator of GSH synthesis, as expected given it's important role in regulating cysteine metabolism (*Sbodio et al., 2018*; *Torrence et al., 2021*), and of the amino acids proline, alanine, and asparagine (*Figure 7B*). Interestingly, aspartate levels increased with Atf4 knockdown in keeping with the role of Atf4 in upregulating *Asns* expression (*Figure 7A*). At the same time, intracellular glutamine levels increased in line with an impairment in GSH synthesis (*Figure 7B*). A comparison of the fold change in amino acids with TCAi, with or without Atf4 silencing, revealed that maintenance of asparagine synthesis was dependent on Atf4-induced *Asns* expression (*Figure 7C*). Knockdown of Atf4 also led to a general dysregulation of intracellular amino acid levels, an increase in cystine (*Figure 7C*) and altered GSH metabolism, such as decreased γ-glutamylcysteine, upon TCAi (*Figure 7—figure supplement 1A,B*). Overall, this work identified a key role for the ISR and Atf4 in regulating the response to TCAi and countering the associated redox and amino acid stress (*Figure 7D*).

## Discussion

Despite the importance of the TCA cycle to cellular metabolism, bioenergetics and redox homeostasis, a comprehensive and unbiased analysis of how mammalian cells respond to TCA cycle dysfunction has not been reported. The ability to perform such analysis is only now possible due to the advent of powerful mass spectrometry and next-generation sequencing technologies that allow us to observe the transcriptome, metabolome, and proteome in greater detail than ever before (*Hasin et al., 2017*). Here, we undertook a multi-modal analysis to elucidate how the TCA cycle operates and integrates with the transcriptome and proteome of the cell. Our analysis revealed significant differences when comparing the metabolic profile of FH or SDH inhibition. These key differences likely arise for several reasons. FH, unlike SDH, is also localised to the cytosol and can also translocate to the nucleus upon DNA damage (*Adam et al., 2013*; *Jiang et al., 2015*). As such, FH loss may give rise to extramitochondrial consequences. SDH is physically associated with the ETC and tethered to the inner mitochondrial membrane. The different distribution of both enzymes within mitochondria is likely to influence their impact mitochondrial bioenergetics and on the overall metabolic profile. A major difference between

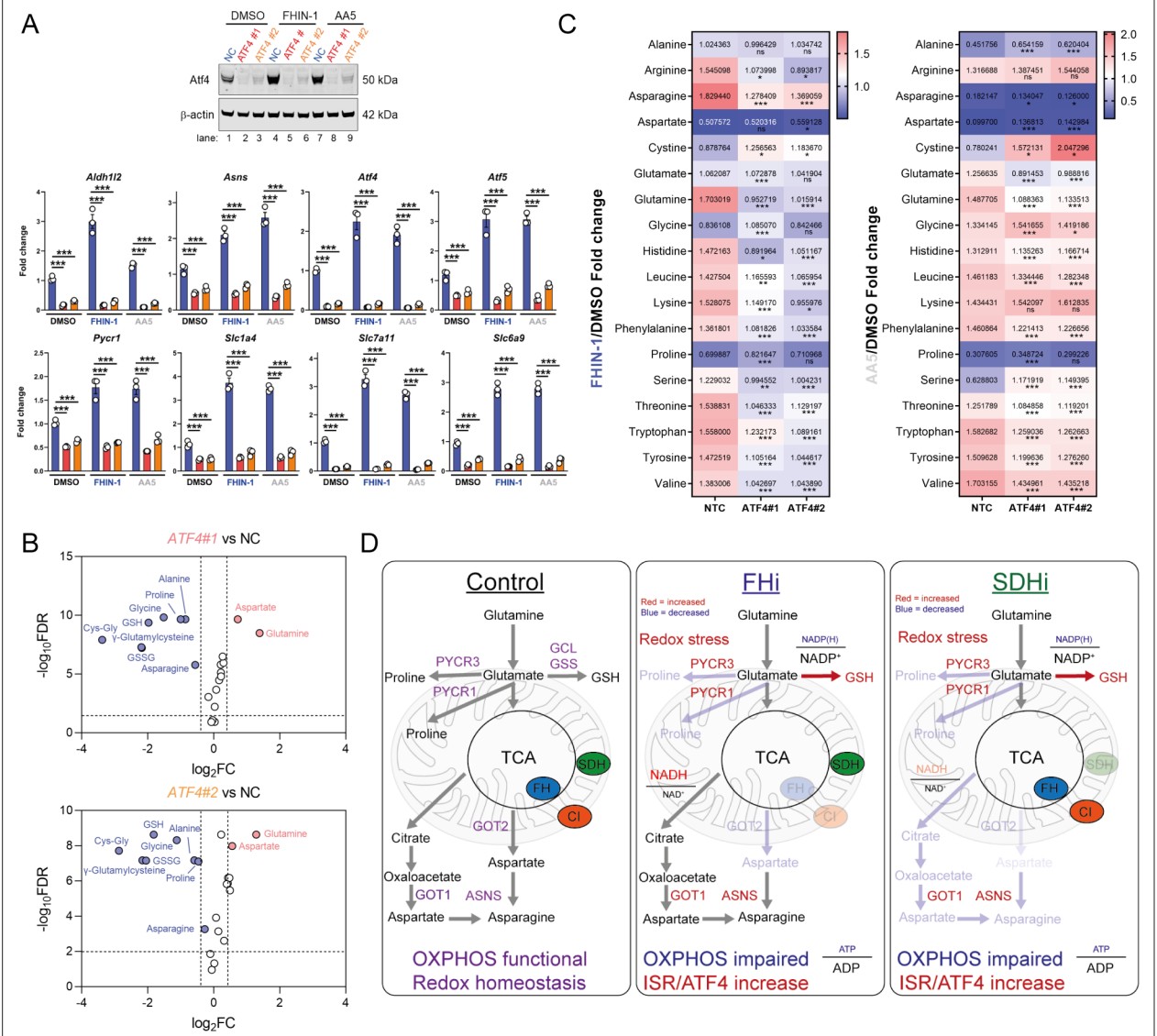

**Figure 7.** Atf4 regulates amino acid and GSH metabolism. (**A**) Western blot analysis of Atf4 and β-actin as loading control (24 hr timepoint) (representative image of 3 independent biological replicates) and qPCR analysis of Atf4-target genes (24 hr timepoint) (n = 3 independent biological replicates) in DMSO-, FHIN-1- and AA5-treated *Fh1^(fl/fl)* cells with or without siRNA-mediated silencing of *Atf4*. Data are mean ± SEM. p value determined by ordinary one-way ANOVA, corrected for multiple comparisons using Tukey statistical hypothesis testing. p < 0.05*; p < 0.01**; p < 0.001***. (**B**) Volcano plot of glutathione-related metabolites and amino acids in Atf4-silenced *Fh1^(fl/fl)* cells versus non-targeting control (NC)- *Fh1^(fl/fl)* cells (48 hr timepoint). (**C**) Heatmap of amino acid levels in FHIN-1- and AA5- versus DMSO-treated *Fh1^(fl/fl)* cells with or without siRNA-mediated silencing of *Atf4* (24 hr timepoint). (**B–C**) (n = 5 independent biological replicates). (**D**) Schematic diagram highlighting key findings of the comparative analysis between acute FHi versus SDHi.

The online version of this article includes the following figure supplement(s) for figure 7:

**Figure supplement 1.** Changes in glutathione metabolism intermediates upon Atf4 silencing and acute TCA cycle inhibition.

FH and SDH loss is the accumulation of fumarate. Fumarate is a mildly electrophilic metabolite that can succinate GSH and protein cysteine residues to form a post-translational modification termed succination, as previously mentioned. In contrast, SDH loss results in a decrease of fumarate and an accumulation of succinate. Unlike fumarate, succinate is not an electrophilic compound that can modify cystine residues and so differences between FH and SDH loss are likely owed to succination, at least in part. Finally, succinate released from cells can bind to the succinate receptor SUCNR1, which is expressed in the kidney (**He et al., 2004**). Autocrine and paracrine ligation of SUCNR1 by high levels

of succinate accumulation and release with SDH loss is likely to alter the metabolic and transcriptional landscape of the cells.

From a metabolic perspective, our analysis also revealed an unexpected link between a functional oxidative TCA cycle, mitochondrial respiration and thiol redox homeostasis with de novo proline biosynthesis. This is reminiscent of the work previously linking appropriate ETC function with aspartate synthesis (*Birsoy et al., 2015*; *Cardaci et al., 2015*; *Sullivan et al., 2015*) and suggests that mitochondrial function is also essential for de novo proline synthesis in certain contexts. Our analysis implicated decreased α-KG siphoning and diminished mitochondrial ATP levels as the mechanism by which the TCA cycle impairs proline synthesis. Recently, two elegant studies revealed that contrary to previous reports, NAD(P)H and not NADH is the primary redox cofactor required for mitochondrial proline synthesis (*Tran et al., 2021*; *Zhu et al., 2021*). Given the drop in the NADPH/NADP$^+$ ratio we observe and the reliance of mitochondrial NADP$^+$ pools on ATP and the NAD kinase NADK2 (*Tran et al., 2021*; *Zhu et al., 2021*), it is tempting to speculate that a combined bioenergetic and reductive power defect plays a role upon TCAi. Furthermore, studies show that Pycr1 is a redox sensitive enzyme that can be modified by reactive electrophilic species (RES) (*Timblin et al., 2021*), while proline synthetic enzymes are oxidative stress-inducible (*Krishnan et al., 2008*) and are also targets of reactive oxygen specie (ROS)-mediated cysteine oxidation, as determined by quantitative redox proteomics of murine tissues in vivo (OxiMouse) (*Xiao et al., 2020*). Given TCAi leads to increased oxidative stress (and succination with FHi) and that direct perturbations of thiol redox homeostasis with mitoCDNB impair proline synthesis, it is also tempting to speculate that ROS or RES could impair this pathway. This could also explain why a common outcome of TCAi is the enhancement of GSH synthesis and why inhibiting GSH synthesis with BSO led to a further decrease in proline levels.

Intriguingly, a novel ribosome profiling technique termed diricore was recently developed to uncover signals of restrictive amino acid availability (*Loayza-Puch et al., 2016*). Application of diricore to a cohort of clear cell renal cell carcinoma (ccRCC) patients, a common renal cancer type associated with severe mitochondrial dysfunction and derived from kidney epithelial cells, uncovered a signal indicating proline restriction (*Loayza-Puch et al., 2016*). The authors also found that ccRCC upregulated PYCR1 in order to compensate for this proline deficit and was a targetable vulnerability in these tumours. As such, our work provides mechanistic insight into how and why proline would be restricted in the context of kidney epithelial cells with mitochondrial dysfunction. ATF4 is likely the transcriptional regulator of the proline synthetic enzymes, Aldh18a1 and Pycr1, in this context. These findings suggest that tumours can compensate for deficiencies in the synthesis of NEAAs linked to mitochondrial function, while tumor-associated stromal cells may also act to provide NEAAs under these conditions. However, identification of these metabolic vulnerabilities may guide therapeutic strategies that utilise diet and pharmacological approaches to treat certain tumour types. While proline and Pycr1 regulation has been linked to Atf4 signaling previously, this was in the context of stem cell differentiation (*D'Aniello et al., 2015*) and not that of mitochondrial dysfunction. Although it has been suggested that the pancreas is the major site of proline production in vivo (*Tran et al., 2021*), the kidney has been proposed to operate as a proline-producing organ under conditions of fasting and with low proline diets (*Watanabe et al., 1999*; *Watanabe et al., 1997*). Given the high density of mitochondria in proximal kidney tubules, it would be tempting to speculate that high rates of oxidative TCA activity could be important to facilitate proline biosynthesis for use by other organs under conditions of fasting or nutrient scarcity.

The importance of mitochondrial retrograde signaling to the cytosol and nucleus, and its regulation of both homeostatic and pathogenic signaling events is only beginning to be appreciated. In this study, we also investigated mitochondrial retrograde signaling events that occur in response to acute TCA cycle dysfunction to better understand the molecular underpinnings of the response. We found that TCAi increased Atf4 levels and triggered a transcriptional response reminiscent of amino deprivation and heme-deficiency. HRI has recently been implicated as the ISR kinase responsible for sensing mitochondrial dysfunction via a newly discovered pathway that involves the proteolytic cleavage of DELE1 by the protease OMA1 (*Fessler et al., 2020*; *Guo et al., 2020*). In contrast, asparagine has also recently been implicated in signaling mitochondrial respiration to Atf4 via GCN2, which detects uncharged aminoacyl tRNAs (*Krall et al., 2021*; *Mick et al., 2020*). Our data likely supports the idea that one or both of these kinases is responsible for sensing TCA cycle dysfunction, however, future investigations are required to fully validate this hypothesis. Our work does however support the idea that reprogramming events are

elicited to support the maintenance of both aspartate and asparagine pools under conditions of mitochondrial dysfunction. In contrast to previous studies, our analysis revealed a surprising degree of metabolic plasticity to this defect in kidney epithelial cells depending on the insult. Cells with acute FHi, but not SDHi, maintained reductive carboxylation and cytosolic aspartate synthesis likely due to decreased Complex I activity and increased NADH/NAD$^+$ ratio as previously described (*Birsoy et al., 2015*; *Mullen et al., 2014*). Unlike acute FHi, chronic FH loss impairs reductive carboxylation in murine kidney epithelial cells (*Frezza et al., 2011*) likely via higher and sustained levels of fumarate-mediated protein succination that affect aconitase function, required for this metabolic re-routing. Acute FHi also led to increased consumption of the branched-chain amino acids (BCAAs) valine and leucine (*Figure 1I*) and an increase in the cytosolic branched chain aminotransferase (BCAT1) (*Figure 5D*), but this was not observed with SDHi. BCAAs can act as an important nitrogen source for the synthesis of cytosolic aspartate (*Mayers et al., 2016*) and so another possibility for the increased capacity for cytosolic aspartate synthesis with FHi is the reprogramming of BCAA catabolism via BCAT1. Importantly, this work also suggests that the relative pool size of aspartate exceeds that of asparagine, as even under conditions of limited aspartate availability, metabolic and transcriptional changes can maintain asparagine levels. This indicates a reprioritization toward maintaining asparagine levels under certain stress conditions despite impaired aspartate synthesis.

In conclusion, both FHi and SDHi impair de novo proline and aspartate synthesis by interrupting oxidative TCA cycle activity and OXPHOS (*Figure 7D*). TCAi also decreases the NADPH/NADP$^+$ ratio to support cystine reduction and promote cytosolic GSH synthesis in order to counteract amino acid and redox stress. A clear upregulation of PSPH, PSAT1, PHGDH, and SHMT1/2 is also observed that may serve to support glutathione synthesis via the provision of glycine. Intriguingly, acute FHi maintains asparagine synthesis via reductive carboxylation, cytosolic aspartate synthesis and an Atf4-induced increase in ASNS. However, SDHi leads to a significant impairment in asparagine synthesis as it fails to engage reductive carboxylation and promote sufficient levels of cytosolic aspartate synthesis. Finally, TCAi triggers activation of the ISR and Atf4 stabilization to communicate an amino-acid-deprived and redox-stressed state to the nucleus. Overall, our work highlights the crucial role that the TCA cycle plays in maintaining metabolic homeostasis and reveals the magnificent plasticity and unique adaptations that may arise in response to metabolic perturbations.

## Limitation of the study

Although comprehensive in its metabolic investigation of acute TCA cycle dysfunction, this study was performed in a proliferating epithelial cell type of kidney origin in vitro. As such, whether these findings extend to different cell types or tissue types will need to be investigated further. While beyond the scope of this study, the precise mechanism(s) by which TCA cycle dysfunction triggers the ISR or how different tissues respond to TCA cycle inhibition in vivo under fed and fasted conditions will also be important to assess in future investigations. Finally, our study used high-nutrient DMEM supplemented with 10% heat inactivated serum. The use of this medium enables careful control of our in vitro experiments and is commonly used for most molecular biology studies. Recently, it has also proved important in revealing specific metabolic dependencies relating to proline (*Tran et al., 2021*; *Zhu et al., 2021*). However, it will be important to assess this response under nutrient scarcity or with different nutrient compositions in the future to reveal other facets of the metabolic response. mitoCDNB is a new tool recently developed to perturb thiol redox homeostasis in mitochondria by depleting GSH and modifying protein cysteines on certain redox enzymes (*Booty et al., 2019*). As such, it is difficult to define whether the effect it has on mitochondrial function and metabolism is due to GSH depletion or direct modification of identified targets. Whether mitoCDNB will modify currently unidentified targets also remains to be determined.

## Materials and methods

**Key resources table**

| Reagent type (species) or resource | Designation | Source or reference | Identifiers | Additional information |
|---|---|---|---|---|
| Cell line (*Mus musculus*) | *Fh1*$^{fl/fl}$ | *Frezza et al., 2011* | *Fh1*$^{fl/fl}$ | Kidney epithelial cell line |
| Cell line (*Mus musculus*) | *Fh1*$^{-/-CL1}$ | *Frezza et al., 2011* | *Fh1*$^{-/-CL1}$ | Kidney epithelial cell line |

*Continued on next page*

*Continued*

| | | | | |
|---|---|---|---|---|
| Cell line (*Mus musculus*) | *Fh1⁻/⁻CL19* | **Frezza et al., 2011** | *Fh1⁻/⁻CL19* | Kidney epithelial cell line |
| Cell line (*Mus musculus*) | *Sdhb^{fl/fl}* | **Cardaci et al., 2015** | *Sdhb^{fl/fl}* | Kidney epithelial cell line |
| Cell line (*Mus musculus*) | *Sdhb⁻/⁻CL5* | **Cardaci et al., 2015** | *Sdhb⁻/⁻CL5* | Kidney epithelial cell line |
| Cell line (*Mus musculus*) | *Sdhb⁻/⁻CL7* | **Cardaci et al., 2015** | *Sdhb⁻/⁻CL7* | Kidney epithelial cell line |
| Chemical compound, drug | Dimethyl sulfoxide (DMSO) | Sigma Aldrich | D8418 | |
| Chemical compound, drug | Fumarate hydratase inhibitor 1 (FHIN-1) | MedChemExpress | Cat. No.: HY-100004 | |
| Chemical compound, drug | Atpenin A5 (AA5) | Abcam | CAS Number: 119509-24-9 | |
| Chemical compound, drug | Thenoyltrifluoroacetone (TTFA) | Sigma Aldrich | T27006 | |
| Chemical compound, drug | L-Buthionine-sulfoximine (BSO) | Sigma Aldrich | B2515 | |
| Chemical compound, drug | ethylGSH (eGSH) | Sigma Aldrich | G1404 | |
| Chemical compound, drug | Oligomycin A | Sigma Aldrich | 75,351 | |
| Chemical compound, drug | Puromycin dihydrochloride | Sigma Aldrich | P9620 | |
| Chemical compound, drug | Antimycin A | Sigma Aldrich | A8674 | |
| Chemical compound, drug | Carbonyl cyanide-p-trifluoromethoxyphenylhydrazone (FCCP) | Sigma Aldrich | C2920 | |
| Chemical compound, drug | Rotenone | Sigma Aldrich | R8875 | |
| Chemical compound, drug | Integrated stress response inhibitor (ISRIB) | Sigma Aldrich | SML0843 | |
| Chemical compound, drug | Glucose | Sigma Aldrich | G8270 | |
| Chemical compound, drug | Glutamine | Sigma Aldrich | G7513 | |
| Chemical compound, drug | Pyruvate | Sigma Aldrich | P2256 | |
| Chemical compound, drug | U-13C-Glutamine | Cambridge Isotopes | CLM-1822-H-PK | |
| Chemical compound, drug | 35S-methionine | PerkinElmer | NEG009L005MC | |
| Chemical compound, drug | mitoCDNB | **Booty et al., 2019** | SML2573* | *Available on Sigma |
| Antibody | puromycin antibody clone 12D10 "(mouse monoclonal)" | Sigma Aldrich | MABE343 | "(1:10,000)" |
| Antibody | anti-Atf4 (D4B8) "(rabbit monoclonal)" | Cell Signalling Technology | Rabbit mAb #11,815 | "(1:800)" |
| Antibody | anti-p-eIF2α (Ser51) (D9G8) "(rabbit monoclonal)" | Cell Signalling Technology | XP Rabbit mAb #3,398 | "(1:1000)" |
| Antibody | anti-β-Actin (13E5) "(rabbit monoclonal)" | Cell Signalling Technology | Rabbit mAb #4,970 | "(1:1000)" |
| Chemical compound, drug | U-13C-Glutamine | Cambridge Isotopes | CLM-1822-H-PK | |
| Chemical compound, drug | 35S-methionine | PerkinElmer | NEG009L005MC | |
| Transfected construct (*Mus musculus*) | Silencer select Non-targeting control (NTC) siRNA | Thermo Scientific | Catalog # 4390843 | |
| Transfected construct (*Mus musculus*) | Silencer select *Atf4* siRNA #1 | Thermo Scientific | Catalog # 4390771 Assay ID s62691 | |
| Transfected construct (*Mus musculus*) | Silencer select*Atf4* siRNA #2 | Thermo Scientific | Catalog # 4390771 Assay ID s62690 | |
| Transfected construct (*Mus musculus*) | Lipofectamine RNAiMAX | Thermo Scientific | Catalog # 13778075 | |

## Drug treatments

All compounds used DMSO as a vehicle except eGSH, which used ultrapure water. All compounds were made up as stock concentrations and diluted 1/1000 – 1/500 in cell culture medium (CCM) prior to treatment. CCM with treatment compounds were vortexed briefly to ensure even distribution of the compounds before adding to appropriate wells or dishes for the indicated timepoints (1, 3, 6, or 24 hr). The final percentage of DMSO in CCM was never more than 0.2%. FHIN-1 (20 µM), AA5 (1 µM) and TTFA (500 µM) treatments for metabolomics, transcriptomics and proteomics were for 24 hr, and also 1 hr for metabolomics, as indicated in figure legends. MitoCDNB (10 µM) treatment for metabolomics was 24 hr Oligomycin (10 µM) treatment for metabolomics was 1 hr. Antimycin A (2 µM) treatment for metabolomics was 24 hr. BSO (500 µM) and eGSH (1 mM) were co-treated with FHIN-1 or AA5 for 24 hr. FHIN-1 (10 µM) and ISRIB (500 nM) was co-treated for western blotting, metabolomics and qPCR for 6 hr or 24 hr as indicated in figure legends.

## Cell culture

*Fh1*-proficient (*Fh1*$^{fl/fl}$) cells and the two *Fh1*-deficient clones (*Fh1*$^{−/−CL1}$ and *Fh1*$^{−/−CL19}$) were obtained, as previously described (*Frezza et al., 2011*; *Sciacovelli et al., 2016*). *Fh1*$^{−/−+pFh-GFP}$ cells were generated from *Fh1*$^{−/−CL1}$ after stable expression of a plasmid carrying either full-length, as previously described (*Sciacovelli et al., 2016*). *Sdhb*-proficient (*Sdhb*$^{fl/fl}$) and two *Sdhb*-deficient clones (*Sdhb*$^{-/-CL5}$ and *Sdhb*$^{-/-CL7}$) were a gift from Prof. Eyal Gottlieb (*Cardaci et al., 2015*). Murine cells were cultured using DMEM (Gibco-41966–029) supplemented with 10% heat-inactivated serum (Gibco-10270–106). Genotyping of cells was assessed as previously described (*Sciacovelli et al., 2016*). All cells were authenticated by short tandem repeat (STR) and routinely checked for mycoplasma contamination. Counting for cell plating and volume measurement were obtained using the CASY cell counter (Omni Life Sciences). Briefly, cells were gently detached using trypsin-EDTA 0.05%, centrifuged at 1500 rpm for 5 min and the cell pellet was re-suspended in fresh media prior to counting.

## LC-MS metabolomics

### Steady-state metabolomics

For steady-state metabolomics, $5 \times 10^4$ cells were plated the day before onto six-well plates (five biological replicates from independently maintained cell lines) and extracted at the appropriate experimental endpoint. Prior to metabolite extraction, cells were counted using CASY cell counter (Omni Life Sciences) using a separate counting plate prepared in parallel and treated exactly like the experimental plate. The counting plate assesses differences in cell number between conditions and the quantity of extraction buffer added is adjusted to this value. This represents the first step in the normalization of the data in our pipeline. For consumption release experiments (CoRe) experiments a Day 0 counting plate was also required to assess proliferation rates. At the experimental endpoint, an aliquot of cell culture conditioned media (CCM) was collected to facilitate CoRe analysis. The remaining media was aspirated off and the cells were washed at room temperature with PBS and placed on a cold bath with dry ice. Metabolite extraction buffer (MES) was added to each well following the proportion $1 \times 10^6$ cells/0.5 ml of buffer. After 10 minutes, the plates were stored at –80 °C freezer and kept overnight. The following day, the extracts were scraped and mixed at 4 °C for 15 min in a thermomixer at 2000 rpm. After final centrifugation at max speed for 20 min at 4 °C, the supernatants were transferred into labeled LC-MS vials.

## Tracing experiments

A total of $5 \times 10^4$ cells were plated onto six-well plate (five biological replicates from independently maintained cell lines for each condition). At the experimental starting point, the medium was replaced with fresh media containing fully labeled U-$^{13}$C-Glutamine (obtained from Cambridge Isotopes Laboratories) and the compound(s) of interest and left on for the duration of the experiment (24 hr).

## Liquid chromatography coupled to mass spectrometry (LC-MS) analysis

HILIC chromatographic separation of metabolites was achieved using a Millipore Sequant ZIC-pHILIC analytical column (5 µm, 2.1 × 150 mm) equipped with a 2.1 × 20 mm guard column (both 5 mm particle size) with a binary solvent system. Solvent A was 20 mM ammonium carbonate, 0.05% ammonium hydroxide; Solvent B was acetonitrile. The column oven and autosampler tray were held at

40 °C and 4 °C, respectively. The chromatographic gradient was run at a flow rate of 0.200 mL/min as follows: 0–2 min: 80% B; 2–17 min: linear gradient from 80% B to 20% B; 17–17.1 min: linear gradient from 20% B to 80% B; 17.1–22.5 min: hold at 80% B. Samples were randomized and analyzed with LC–MS in a blinded manner with an injection volume was 5 µl. Pooled samples were generated from an equal mixture of all individual samples and analyzed interspersed at regular intervals within sample sequence as a quality control.

Metabolites were measured with a Thermo Scientific Q Exactive Hybrid Quadrupole-Orbitrap Mass spectrometer (HRMS) coupled to a Dionex Ultimate 3000 UHPLC. The mass spectrometer was operated in full-scan, polarity-switching mode, with the spray voltage set to +4.5 kV/–3.5 kV, the heated capillary held at 320 °C, and the auxiliary gas heater held at 280 °C. The sheath gas flow was set to 25 units, the auxiliary gas flow was set to 15 units, and the sweep gas flow was set to 0 unit. HRMS data acquisition was performed in a range of $m/z$ = 70–900, with the resolution set at 70,000, the AGC target at $1 \times 10^6$, and the maximum injection time (Max IT) at 120ms. Metabolite identities were confirmed using two parameters: (1) precursor ion m/z was matched within 5 ppm of theoretical mass predicted by the chemical formula; (2) the retention time of metabolites was within 5% of the retention time of a purified standard run with the same chromatographic method. Chromatogram review and peak area integration were performed using the Thermo Fisher software Tracefinder 5.0 and the peak area for each detected metabolite was normalized against the total ion count (TIC) of that sample to correct any variations introduced from sample handling through instrument analysis. The normalized areas were used as variables for further statistical data analysis.

For $^{13}$C-tracing analysis, the theoretical masses of $^{13}$C isotopes were calculated and added to a library of predicted isotopes. These masses were then searched with a five ppm tolerance and integrated only if the peak apex showed less than 1% difference in retention time from the [U-$^{12}$C] monoisotopic mass in the same chromatogram. After analysis of the raw data, natural isotope abundances were corrected using the AccuCor algorithm (https://github.com/lparsons/accucor) (*Su et al., 2017*).

## RNA sequencing

A total of $5 \times 10^4$ cells were plated onto three replicate 6-cm dishes before the extraction and treated as indicated. RNA isolation was carried using RNeasy kit (Qiagen) following manufacturer's suggestions and eluted RNA was purified using RNA Clean & Concentrator Kits (Zymo Research). RNA-seq samples libraries were prepared using TruSeq Stranded mRNA (Illumina) following the manufacturer's description. For the sequencing, the NextSeq 75 cycle high output kit (Illumina) was used and samples spiked in with 1% PhiX. The samples were run using NextSeq 500 sequencer (Illumina). Differential Gene Expression Analysis was done using the counted reads and the R package edgeR version 3.26.5 (R version 3.6.1) for the pairwise comparisons.

## Cellular fractionation and mitochondrial isolation

Cells were grown to confluency on 2 × 150 mm dishes per indicated treatments. At the experimental endpoint, cells were washed 4 x with ice cold 1 X PBS on ice. All excess PBS was removed after the final wash. 500 µl of ice-cold STE buffer (250 mM sucrose, 5 mM Tris, 1 mM EGTA, pH 7.4, 4 °C) was added to each dish and cells were scraped and transferred to a pre-cooled 7 ml dounce homogenizer and cells homogenized with ~100 strokes of a tight-fitting pestle. The homogenate was passed through a 30 G needle ten times and split into 1.5 mL centrifuge tubes. Cells were then centrifuged at 3000 x g for 3 min at 4 °C. The supernatant was transferred to a new centrifuge tube and centrifuged as before. The supernatant was transferred to a new tube and centrifuge at 11,000 x g for 5 min at 4 °C. The supernatant 'cytosol' fraction was taken and placed in a new centrifuge tube prior to metabolomic extraction as above. The remaining pellet was resuspended in 1 mL STE buffer and centrifuged again for 11,000 x g for 5 min at 4 °C to generate the 'mitochondrial' fraction. All supernatant was removed prior to metabolite extraction.

## Proteomic analysis
### Sample preparation

A total of $5 \times 10^4$ cells were plated onto five replicate 10-cm dishes, grown to confluence and treated as indicated. At the experimental endpoint, cells were washed with PBS on ice and centrifuged at 1500 rpm for 5 min at 4 °C and frozen at –80 °C. Cell pellets were lysed, reduced and alkylated in

100 µl of 6 M Gu-HCl, 200 mM Tris-HCl pH 8.5, 1 mM TCEP, 1.5 mM Chloroacetamide by probe soni-cation and heating to 95 °C for 5 min. Protein concentration was measured by a Bradford assay and initially digested with LysC (Wako) with an enzyme to substrate ratio of 1/200 for 4 hr at 37 °C. Subsequently, the samples were diluted 10fold with water and digested with porcine trypsin (Promega) at 37 °C overnight. Samples were acidified to 1% TFA, cleared by centrifugation (16,000 g at RT) and approximately 20 µg of the sample was desalted using a Stage-tip. Eluted peptides were lyophilized, resuspended in 0.1% TFA/water and the peptide concentration was measured by A280 on a nanodrop instrument (Thermo). The sample was diluted to 1 µg/ 5 µl for subsequent analysis.

## Mass spectrometry analysis

The tryptic peptides were analyzed on a Fusion Lumos mass spectrometer connected to an Ultimate Ultra3000 chromatography system (both Thermo Scientific, Germany) incorporating an autosampler. Five µL of the tryptic peptides, for each sample, was loaded on an Aurora column (Ionoptiks, Melbourne Australia) and separated by an increasing acetonitrile gradient, using a 150-min reverse-phase gradient (from 3%–40% Acetonitrile) at a flow rate of 400 nL/min. The mass spectrometer was operated in positive ion mode with a capillary temperature of 220 °C, with a potential of 1500 V applied to the column. Data were acquired with the mass spectrometer operating in automatic data-dependent switching mode, with MS resolution of 240 k, with a cycle time of 1 s and MS/MS HCD fragmentation/analysis performed in the ion trap. Mass spectra were analyzed using the MaxQuant Software package in biological triplicate. Label-free quantitation was performed using MaxQuant. All the samples were analyzed as biological triplicates.

## Data analysis

Data were analyzed using the MaxQuant software package. Raw data files were searched against a human database (Uniprot *Homo sapiens*), using a mass accuracy of 42.5 ppm and 0.01 false discovery rate (FDR) at both peptide and protein level. Every single file was considered as separate in the experimental design; the replicates of each condition were grouped for the subsequent statistical analysis. Carbamidomethylation was specified as fixed modification while methionine oxidation and acetylation of protein N-termini were specified as variable. Subsequently, missing values were replaced by a normal distribution (1.8 π shifted with a distribution of 0.3 π) in order to allow the following statistical analysis. Results were cleaned for reverse and contaminants and a list of significant changes was determined based on average ratio and t-test. LFQ-analyst was used to perform differential expression analysis after pre-processing with MaxQuant (*Shah et al., 2020*).

## Oxygen consumption rate (OCR) measurements

Cellular respiration (OCR), as part of the Cell Mito Stress test, was measured using the real-time flux analyzer XF-24e Seahorse (Agilent). In brief, $6 \times 10^4$ cells were plated onto the instrument cell plate 24 hr before the experiment in complete DMEM with 10% FBS (five replicate wells for each condition). The following day cells were treated as indicated for 24 hr. At the treatment endpoint, the medium was replaced with DMEM pH 7.4 (Agilent) supplemented with 25 mM glucose, 2.5 mM glutamine and 1 mM pyruvate prior to Cell Mito Stress Test analysis according to manufacturer's instructions. Cells were treated with 2.5 µM Oligomycin, 0.5 µM FCCP and 0.5 µM Antimycin A/Rotenone to assess different respiration parameters.

## RNA extraction and real-time quantitative (QPCR)

A total of $5 \times 10^4$ cells were plated onto 6-well plates, left to adhere overnight and treated for indicated times. At experimental endpoint, cells were washed in 1 X PBS and then RNA was extracted using RNeasy kit (Qiagen) following the manufacturer's instructions. RNA was eluted in water and then quantified using a Nanodrop (ThermoFisher Scientific). One µg of RNA was reverse- transcribed using High capacity cDNA Reverse Transcription kit (Applied Biosystems). For real-time qPCR, cDNA was run using Fast SYBR Green Master Mix (Applied Biosystems) according to manufacturer's instructions and primers were designed for genes of interest (see below) using primer-BLAST. *Actb* was used as the endogenous control. qPCR experiments were run on a 384-well QuantStudio Real-Time PCR system (ThermoFisher Scientific).

## qPCR primers

| Gene | Forward (5'–3') | Reverse (5'–3') |
|---|---|---|
| Actb | TTTGCAGCTCCTTCGTTGC | TTCCCACCATCACACCCTGG |
| Atf4 | CCTCCCGCAGGGCTTG | GATTTCGTGAAGAGCGCCAT |
| Atf5 | TGGCTCGTAGACTATGGGAAAC | ATCAGAGAAGCCGTCACCTGC |
| Slc7a11 | CTGGGTGGAACTGCTCGTAATA | ATCACCACAGTGATGCCCACA |
| Slc1a4 | GTGGTTGCCGCATTCACTAC | AGGGCAAAAAGGACGAGACC |
| Slc6a9 | CGGGAGGCTGATGCAACTT | GGCACAGCACCATTCAACATC |
| Ddit3 | GCAGCGACAGAGCCAGAATA | CCAGGTTCTGCTTTCAGGTG |
| Aldh1l2 | GTTTTCTGGGCAGATGATGGT | TCCCAAGAAACCTCAGCGT |
| Asns | ACGACAGTTCGGGCATCTG | GGAAGGAGCCTTGTGGAAATA |
| Pycr1 | ACCCGAATATCCACGCTTTCT | TTGTGGGGTGTCAGGTTCAC |
| Lonp1 | CCTGTGTTCCCGCGCTTTAT | CGTCCGACTCATTGTTGTCA |

## Puromycin assay

A total of $5 \times 10^4$ cells were plated onto 12-well plates, left to adhere overnight and treated with FHIN-1, AA5 and TTFA for indicated times (1, 3, or 6 hr). For the last 15 min of the treatment time, 20 µg/mL puromycin dihydrochloride (Sigma) was added to the well prior to extraction. Puromycin incorporation into neosynthesized proteins was assessd with western blotting and an anti-puromycin antibody (Sigma).

## Western blotting

A total of $5 \times 10^4$ cells were plated onto 6- or 12-well plates, left to adhere overnight and treated for indicated times. At the experimental endpoint, cells were counted on a parallel counting plate using a CASY counter, washed in 1 X PBS and then lysed on ice with the appropriate volume (100 µL per 100,000 cells) of 4 X Bolt Loading buffer (Thermo Scientific) diluted to 1 X with RIPA buffer (150 mM NaCl, 1% NP-40, Sodium deoxycholate (DOC) 0.5%, sodium dodecyl phosphate (SDS) 0.1%, 25 mM Tris) supplemented with protease and phosphates inhibitors (Protease inhibitor cocktail, Phosphatase inhibitor cocktail 2/3) (Sigma-Aldrich) and containing 4% β-mercaptoethanol. Protein samples were then heated at 70 °C for 10 min, briefly centrifuged and stored at –20 °C for future use. Samples were loaded onto 4–12% Bis-Tris Bolt gradient gels and run at 160 V constant for 1 hr in Bolt MES 1 X running buffer (Thermo Scientific). Dry transfer of the proteins onto a nitrocellulose membrane was done using iBLOT2 (Thermo Scientific) for 12 min at 20 V. Membranes were incubated in blocking buffer for 1 hr (either 5% BSA or 5% milk in TBS 1X + 0.01% Tween-20, TBST 1 X). Primary antibodies were incubated in blocking buffer ON at 4 °C under gentle agitation. On the following day, the membranes were washed three times in TBST 1 X for 5 min and then secondary antibodies (conjugated with 680 or 800 nm fluorophores) (Li-Cor) incubated for 1 hr at room temperature at 1:2000 dilution in blocking buffer. Images were acquired and quantified using Image Studio lite 5.2 (Li-Cor) on Odyssey CLx instrument (Li-Cor).

## RNAi transfection

A total of $5 \times 10^4$ cells were plated onto six-well plates and left to adhere overnight. On the following day, plates were transfected with indicated Silencer select siRNAs (10 nM) complexed with Lipofectamine RNAiMAX and Opti-MEM medium according to manufacturer's instructions. After 24 hr incubation, cells were treated as indicated prior to downstream processing for metabolomics, western blot or qPCR.

## $^{35}$S-methionine labeling of mitochondrial translation products

In order to label newly synthesised mitochondrially expressed proteins, the previously published protocol was used (*Pearce et al., 2017*). Briefly, cells at approximately 80% confluency were incubated

in methionine/cysteine-free medium for 10 min before the medium was replaced with methionine/cysteine-free medium containing 10% dialysed FCS and emetine dihydrochloride (100 µg/ml) to inhibit cytosolic translation. Following a 20-min incubation, 120 µCi/ml of [$^{35}$S]-methionine was added and the cells were incubated for 30 min. After washing with 1 X PBS, cells were lysed, and 30 µg of protein was loaded on 10–20% Tris- glycine SDS-PAGE gels. Dried gels were visualized with a PhosphorImager system.

## Statistical analysis

Graphs were generated using Graphpad prism 9.0 software, which was used to perform most statistical analysis. Metaboanalyst 5.0 was used to analyze metabolomics data from *Figure 1A–C*. ORA analysis of the significant hits from RNAseq and proteomics used Enrichr (*Chen et al., 2013*). RNA seq cut-offs were set to $\log_2$FC of 1 and an FDR < 0.05. Proteomics cut offs were set to $\log_2$FC of 0.5 and FDR < 0.05. GSEA analysis of RNAseq was performed using the Broad Institutes GSEA 4.1.0 (*Subramanian et al., 2005*).

## Acknowledgements

We thank Cambridge Genomic Services (Department of Pathology, University of Cambridge) especially Dr Alexandria Karcanias and Dr Julien Bauer for the RNA-seq library preparation, sequencing and differential expression analysis. We thank Prof. David Ron for helpful discussions. We also thank all the Frezza lab members for all their helpful discussions and insights and Prof. Richard Hartley who originally synthesized mitoCDNB. M.M and C.A.P are funded by Medical Research Council (MRC) core grant to the MRC Mitochondrial Biology Unit (MC_UU_00015/4).

## Additional information

### Funding

| Funder | Grant reference number | Author |
| --- | --- | --- |
| Medical Research Council | MRC_MC_UU_12022/6. | Christian Frezza |
| H2020 European Research Council | ERC819920 | Dylan Gerard Ryan Christian Frezza |

The funders had no role in study design, data collection and interpretation, or the decision to submit the work for publication.

### Author contributions

Dylan Gerard Ryan, Conceptualization, Data curation, Formal analysis, Investigation, Methodology, Validation, Visualization, Writing – original draft, Writing – review and editing; Ming Yang, Hiran A Prag, Giovanny Rodriguez Blanco, Efterpi Nikitopoulou, Marc Segarra-Mondejar, Christopher A Powell, Tim Young, Nils Burger, Jan Lj Miljkovic, Investigation, Writing – review and editing; Michal Minczuk, Michael P Murphy, Supervision, Writing – review and editing; Alex von Kriegsheim, Investigation, Supervision, Writing – review and editing; Christian Frezza, Conceptualization, Funding acquisition, Project administration, Supervision, Writing – original draft, Writing – review and editing

### Author ORCIDs

Dylan Gerard Ryan  http://orcid.org/0000-0003-4553-9192
Christopher A Powell  http://orcid.org/0000-0001-7501-0586
Tim Young  http://orcid.org/0000-0002-1831-3473
Michael P Murphy  http://orcid.org/0000-0003-1115-9618
Christian Frezza  http://orcid.org/0000-0002-3293-7397

### Decision letter and Author response

Decision letter https://doi.org/10.7554/eLife.72593.sa1
Author response https://doi.org/10.7554/eLife.72593.sa2

## Additional files

### Supplementary files
• Transparent reporting form

### Data availability
All the transcriptomics. proteomics and uncropped blots data have been deposited in Dryad.

The following dataset was generated:

| Author(s) | Year | Dataset title | Dataset URL | Database and Identifier |
|---|---|---|---|---|
| Ryan D | 2022 | Disruption of the TCA cycle reveals an ATF4-mediated integration of redox and amino acid metabolism | https://doi.org/10.5061/dryad.9ghx3ffjz | Dryad Digital Repository, 10.5061/dryad.9ghx3ffjz |

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
