## [Editor Report]

Ryan et al., compare the effects of acute FH or SDH inhibition with genetic ablation of FH or SDH in kidney epithelial cells. The consider how each intervention affects metabolite levels, alters the fate of labeled nutrients, and how it influences the transcriptome and proteome. This includes showing that disrupting the TCA cycle has a large effect on amino acid metabolism, and activates a stress response to maintain redox and amino acid homeostasis.

---

## [Decision Letter]

**Decision letter after peer review:**

Thank you for submitting your article "Disruption of the TCA cycle reveals an ATF4-dependent integration of redox and amino acid metabolism" for consideration by *eLife*. Your article has been reviewed by 3 peer reviewers, one of whom is a member of our Board of Reviewing Editors, and the evaluation has been overseen by a Senior Editor. The following individuals involved in review of your submission have agreed to reveal their identity: Daniel Tennant (Reviewer #2); Christian Metallo (Reviewer #3).

The reviewers have discussed their reviews with one another, and the Reviewing Editor has drafted this letter to help you prepare a revised submission.

Essential revisions:

The reviewers agreed that this work is the first to comprehensively look at the differences, rather than the similarities, between FH and SDH disruption, although agreed that some additional discussion of what might drive those differences is warranted. The reviewers each made specific comments, and agreed that none necessarily require additional experiments, but to enhance the impact of the paper I have copied the full reviewer comments below. I also ask you pay particular attention to these points when adding additional discussion in your revised manuscript.

1. Please clarify how data were normalized (Reviewer 1 point 1).

2. Please discuss what may account for differences observed between SDH and FH loss.

3. Please check all figures to ensure data are accessible (i.e. not too small to evaluate) and consider the points raised by Reviewer 2 regarding data presentation.

4. Please clarify the timing of all treatments and discuss limitations of the compounds used (Reviewer 2).

5. Please consider the various suggestions raised by Reviewer 2 in their extensive review.

6. Please discuss that NEAA may be available from other cell types in tumors with SDH or FH loss.

*Reviewer #1 (Recommendations for the authors):*

As noted above – the authors should address:

1. Exactly how each perturbation affects cell proliferation is not clear. This should be considered, as whether some of the differences are a result in changes in growth or proliferation rate is possible, and will affect how they normalize their data.

2. It is unclear why FH loss is different than SDH loss, and it is also somewhat surprising that the effects of acute and chronic loss of either enzyme are not that different. While explaining this is too much to ask, some additional speculation might be warranted.

3. The increase in glutathione and GSSG is interpreted as a consequence of increased oxidative stress, but that will not necessarily affect total levels.

4. The text in the Figure S1 PCA plots have legends is too small to read. This should be corrected.

*Reviewer #2 (Recommendations for the authors):*

Overall a very elegant manuscript that pulls together complex datasets to identify a coherent signalling mechanism between dysfunctional mitochondrial metabolism and a nuclear transcriptional response. I feel that there are a few areas in which the manuscript could be improved from both the technical and 'readability' perspectives, but overall it is of a very high standard.

1. The idea of a single figure legend for multiple graphs is often excellent. However, due to the large number of colours and treatment conditions, I found it a bit difficult to interpret some of the figures with this layout. For example, where some graphs are MIDs and others are treatments, the single legend meant it took longer to interpret (see Figure 2B, C and E for example). The size of the legend also meant that the large numbers of colours for MIDs could be tricky to interpret on a screen (see Figure 1B – the orange and pink shown in the legend were almost indistinguishable on the screen and when printed out). Some work on the presentation of the data in this way would really aid the reader interpreting some of the metabolic results.

2. The length of treatment with compounds is not always clear – I was unable to find details of the length of oligomycin or BSO treatments used, as examples.

3. In the initial metabolomics screen (Figure 1, S1), I was unable to find succinicGSH in the heatmap (S1E), while it was reported in the graphed data next to it. If it is there, could this be made clearer – perhaps highlighting in bold those metabolites that the authors refer to?

4. As highlighted in the public review, I'm not aware of the mitoCDNB being shown not to interfere with other aspects of mitochondrial metabolism, such as ETC activity. This would also be predicted to have a similar effect on the metabolites shown, based on other evidence in this manuscript. One way of addressing this might be to confirm that respiration continues after mitoCDNB treatment. Short-term BSO treatment has also previously been shown to affect the cytosolic glutathione pool, while the mitochondrial pool is slower to turn over. This should not have the same metabolic dysfunctional effect as longer BSO treatment that depletes total glutathione. Perhaps therefore repeating the BSO experiment shown in Figure 4 with a more acute treatment might be another approach?

5. In Figure 2C, while aspartate m+4 is 20% or less of the untreated levels, asparagine m+4 is closer to 60%. This effect is also apparent in the m+3 isotopomer comparison (Figure S2C) – while aspartate is reduced to around 40% of levels, asparagine is unchanged. Given the significant (2 orders of magnitude) difference in concentration between these metabolites, is this a reprioritisation of aspartate fate? This is also apparent in Figure 4C, in which BSO treatment has a substantial effect on asparagine concentrations, while aspartate is comparatively unaffected.

6. Conventionally, the pyridine nucleotide ratios are either NAD+:NADH and NADP+:NADPH, or as a generalised form NAD(P)}:NAD(P)H. In some cases, the authors are using a mixed ratio – NAD(P)H:NADP+ or similar. It isn't clear why this is important, or whether this is just an oversight? Indeed, given that the Tran et al., (2021) paper showed that NADK2 is required for P5C synthesis, can we now be more certain that ALDH18A1 is most likely to utilise NADPH under physiological conditions?

7. In Figure 4A, the BSO values appear to be close to 0, or not detected. However, as it stands this could be clearer – could the authors use another method – either split axis or log scale, or highlight 'ND' if it was below the analytical noise?

8. Could the authors comment on the possibility that some of the metabolite rescue observed after eGSH treatment could be through recycling of the compound itself into metabolic pools – e.g. glutamate? (Figure S4B).

9. In Figure 4G, use of oligomycin did not alter the cellular NADH:NAD+ ratio. This is a non-obvious result, as it could be expected to change, depending on the time for which oligomycin treatment was used. Could the authors comment on this in the text, or show how the cells are compensating with significantly increased lactate synthesis?

10. From line 266 onwards, the authors suggest that reduced ATP and NADPH availability is the likely explanation for the results observed. However, the authors did not report a change in the NADPH:NADP+ ratio (Figure 4G), so based on the evidence provided, the drop in ATP could be considered sufficient for their hypothesis.

11. In the proteomic analysis, the authors identify a significant change in a significant number of enzymes involved in amino acid metabolism. The change in ASNS, for example was a clearly consistent and significant change across treatments. However, the authors also specifically mention PYCR3 as being altered. While this is the case, there are others that appear more consistently upregulated but not mentioned. One example of this is PSPH/PSAT1/PHGDH/SHMT_1/2_, which are all increased in both FHi and SDHi conditions, and form a fairly coherent metabolic network that is highly compelling. As the authors will be highly aware, the provision of glycine for glutathione synthesis is likely to rely on these enzymes.

*Reviewer #3 (Recommendations for the authors):*

Nice work.

The key limitation I see is: the impact of mito defects on NEAA synthesis is well-established and clear, but this severe metabolic dependence is also a function of in vitro growth conditions. I see the point in "limitations" but it is worth explaining to the reader that NEAAs can be provided by other cell types in the context of tumor growth.

Very nice data figures. I like the presentation of both signal and fractional labeling, which makes the findings/conclusions very easy to review and interpret!

---

## [Author Response]

Reviewer #1 (Recommendations for the authors):As noted above – the authors should address:1. Exactly how each perturbation affects cell proliferation is not clear. This should be considered, as whether some of the differences are a result in changes in growth or proliferation rate is possible, and will affect how they normalize their data.

We thank the referee for raising a critical point and allowing us to clarify how we normalize metabolomics experiments. All the metabolomics data takes into consideration the cell number. Indeed, prior to metabolite extraction, cells are counted using a separate counting plate prepared in parallel and treated exactly like the experimental plate. In this way, differences in cell number are accounted for and the efficiency of metabolic extraction is preserved. Consumption release (CoRe) metabolomics also takes into consideration the proliferation rate during normalization since we normalise data to the final cell number. We have expanded the description in the relevant sections in the Methods.

2. It is unclear why FH loss is different than SDH loss, and it is also somewhat surprising that the effects of acute and chronic loss of either enzyme are not that different. While explaining this is too much to ask, some additional speculation might be warranted.

We postulate that FH loss is different to SDH loss for several reasons:

A. FH is localized to mitochondria (specifically in the mitochondrial matrix) and the cytosol (cytFH). cytFH can translocate to the nucleus to regulate the DNA damage response (PMID: 26237645). In contrast, the SDH complex is only localized to mitochondria. As such, a loss of FH function is likely to have mitochondrial and extramitochondrial consequences.

B. SDH is also the only TCA cycle enzyme that's physically associated with the electron transport chain (ETC) and tethered to the inner mitochondrial membrane, where it also regulates the ubiquinone pool. The different distribution of both enzymes within mitochondria is likely to influence their impact mitochondrial bioenergetics and on the overall metabolic profile.

C. A major difference between FH and SDH loss is the accumulation of fumarate. As discussed in the manuscript, fumarate is a mildly electrophilic metabolite that can succinate GSH and protein cysteine residues to form a post-translational modification termed succination. Fumarate-mediated succination is known to impair iron-sulphur cluster metabolism and perturb aconitase and Complex I function (PMID: 29069586). This is just one example of how succination can affect cellular function. In contrast, SDH loss results in a decrease of fumarate and an accumulation of succinate.Unlike fumarate, succinate is not an electrophilic compound that can modify cystine residues, and so differences between FH and SDH loss are likely owed to succination, at least in part.

D. While it hasn't been investigated in this study, succinate released from cells can bind to the succinate receptor SUCNR1, which is expressed in the kidney (PMID: 21803970). Autocrine and paracrine ligation of SUCNR1 by high levels succinate accumulation and release is likely to alter the metabolic and transcriptional landscape of the cells.

Based on these observations, we also argue that the effect of acute and chronic enzyme inhibition is expectedly different regarding how the key metabolic and signalling hallmarks of FH and SDH loss develop and interact with each other over time. This hypothesis is part of work currently undergoing in our laboratory. For example, chronic SDH loss led to a significant increase in 20 metabolites however, acute SDH inhibition with TTFA and AA5 led to an increase in 60 and 50 metabolites, respectively. Chronic FH loss also led to a significant increase in 92 metabolites, whereas acute FH inhibition led to a significant increase in 49. The fact that only 2 metabolites overlap between all conditions indicates apparent differences between the loss of both enzymes on the metabolome and whether the loss is acute or chronic in nature. There are also notable differences between chronic FH loss and acute FH inhibition in relation to reductive carboxylation. Chronic FH loss triggers a higher accumulation of fumarate and succination of aconitase that impairs reductive carboxylation (PMID: 21849978); however, acute inhibition facilitates reductive carboxylation (Figure S2), likely due to lower levels of succination given the acute treatment. As such, we feel there are notable differences in the metabolite profiles and rewiring events associated with acute versus chronic enzyme inhibition. We have discussed these important points in the Discussion section of the manuscript.

3. The increase in glutathione and GSSG is interpreted as a consequence of increased oxidative stress, but that will not necessarily affect total levels.

We agree that oxidative stress will not necessarily affect total glutathione levels and this finding is likely a time-dependent phenomenon due to persistent redox signalling. In this instance, the alterations in total glutathione levels are likely linked to transcriptional and post-transcriptional changes in GSH biosynthetic enzymes and the observed metabolic reprogramming. While ATF4 regulates the glutathione redox state and glutathione levels, it's not entirely clear if it is solely responsible for increasing total glutathione levels with TCA cycle inhibition. One possibility is that there is simultaneous activation of the transcription factor NRF2, which is a crucial regulator of glutathione synthesis and is known to be regulated by FH loss and succination (PMID: 22014567) and reactive oxygen species (ROS). ATF4 and NRF2 may cooperate to transactivate glutathione-related metabolic enzymes upon TCA cycle inhibition, as previously reported in other contexts (PMID: 23618921). Further investigation of the crosstalk between these two transcription factors is warranted in this context.

4. The text in the Figure S1 PCA plots have legends is too small to read. This should be corrected.

We thank you for this note and apologize for this oversight. We have now corrected the figure legends for the PCA plots.

Reviewer #2 (Recommendations for the authors):Overall a very elegant manuscript that pulls together complex datasets to identify a coherent signalling mechanism between dysfunctional mitochondrial metabolism and a nuclear transcriptional response. I feel that there are a few areas in which the manuscript could be improved from both the technical and 'readability' perspectives, but overall it is of a very high standard.

We thank the referee for the positive feedback on our work

1. The idea of a single figure legend for multiple graphs is often excellent. However, due to the large number of colours and treatment conditions, I found it a bit difficult to interpret some of the figures with this layout. For example, where some graphs are MIDs and others are treatments, the single legend meant it took longer to interpret (see Figure 2B, C and E for example). The size of the legend also meant that the large numbers of colours for MIDs could be tricky to interpret on a screen (see Figure 1B – the orange and pink shown in the legend were almost indistinguishable on the screen and when printed out). Some work on the presentation of the data in this way would really aid the reader interpreting some of the metabolic results.

We thank the reviewer for this constructive feedback, and we have now amended the figure legends separating the MIDs and treatment. We have also increased their size to make them more readable and for data interpretation.

2. The length of treatment with compounds is not always clear – I was unable to find details of the length of oligomycin or BSO treatments used, as examples.

We thank the reviewer for the feedback. All treatment timepoints were in the figure legends, but we have now also included the treatment times in the methods and figure legends. We have also made the timepoints used more evident in the figure legends where possible.

3. In the initial metabolomics screen (Figure 1, S1), I was unable to find succinicGSH in the heatmap (S1E), while it was reported in the graphed data next to it. If it is there, could this be made clearer – perhaps highlighting in bold those metabolites that the authors refer to?

The referee raises a valid point here. The heatmaps provided include the 'Top 50' metabolites ranked by ANOVA, and in this instance, succinicGSH was excluded in this visualization due to this cut-off. For consistency, we opted to stick with Top 50 as opposed to changing the number on display, which can become quite unreadable and dense if increased further.

4. As highlighted in the public review, I'm not aware of the mitoCDNB being shown not to interfere with other aspects of mitochondrial metabolism, such as ETC activity. This would also be predicted to have a similar effect on the metabolites shown, based on other evidence in this manuscript. One way of addressing this might be to confirm that respiration continues after mitoCDNB treatment. Short-term BSO treatment has also previously been shown to affect the cytosolic glutathione pool, while the mitochondrial pool is slower to turn over. This should not have the same metabolic dysfunctional effect as longer BSO treatment that depletes total glutathione. Perhaps therefore repeating the BSO experiment shown in Figure 4 with a more acute treatment might be another approach?

We would like to thank the author for their insight regarding mitoCDNB and the potential caveats of this compound, which we also pondered. mitoCDNB does modify reactive thiols in some mitochondrial enzymes, including TrxR2 and peroxiredoxin 3. This compound may react with other thiols on components of the electron transport chain directly. Indeed, although mitoCDNB was reported to have minimal effect on respiration in a different cell type (PMID: 30713096), the decrease in aspartate level suggests that there is respiration impairment upon treatment with mitoCDNB. To provide an overview of the possible effect of this compound, we have now updated Figure S3 to show the ATP/ADP ratio, which is unchanged, and NADH/NAD^+^ and lactate/pyruvate ratios, which are both increased. The NADH/NAD^+^ ratio does suggest an impairment with Complex I. Given the importance of glutathione in regulating ROS and iron-sulphur clusters (PMID: 32854270; https://www.nature.com/articles/s41586-021-04025-w#Sec4), it is likely that its depletion and perturbation of thiol redox homeostasis will also impair complex I function indirectly. As such, it is difficult to dissect whether the effect of mitoCDNB is mediated by impairment of thiol redox homeostasis or direct inhibition of complex I or the ETC. We have mentioned this caveat in the limitations of the study.

We also thank the reviewer for the suggestion with the BSO treatments and we apologize that the treatment strategy was not clear in the manuscript. Our TCA cycle inhibitors were co-treated with BSO simultaneously for 24 h. We reasoned that intervention represents a more acute inhibition of cytosolic GSH synthesis at the time of TCA cycle inhibition, as opposed to a longer-term depletion of glutathione pools to first deplete both cytosolic and mitochondrial GSH pools. Mitochondrial GSH pools would be completely intact at the time of TCA cycle inhibition given the slower turnover and this was the reason we opted for the co-treatment as opposed to a pre-treatment with BSO.

5. In Figure 2C, while aspartate m+4 is 20% or less of the untreated levels, asparagine m+4 is closer to 60%. This effect is also apparent in the m+3 isotopomer comparison (Figure S2C) – while aspartate is reduced to around 40% of levels, asparagine is unchanged. Given the significant (2 orders of magnitude) difference in concentration between these metabolites, is this a reprioritisation of aspartate fate? This is also apparent in Figure 4C, in which BSO treatment has a substantial effect on asparagine concentrations, while aspartate is comparatively unaffected.

We thank the reviewer for highlighting this important finding, which we did not elaborate sufficiently due to space constraints. This observation likely reflects the different size in the aspartate and asparagine pools and the increase in ASNS and cytosolic aspartate synthesis upon TCA cycle inhibition. We speculate that a smaller asparagine pool makes this aminoacid a sensitive sensor of the nutrient state of the cell and of asparate levels. In addition, this finding speak for a reprioritization, as adeptly put by the reviewer, of aspartate for asparagine synthesis, despite a depletion of aspartate under stress conditions. This finding is especially noticeable with acute FHi. We have included a mention of this important point in the Discussion section of the manuscript.

6. Conventionally, the pyridine nucleotide ratios are either NAD+:NADH and NADP+:NADPH, or as a generalised form NAD(P)}:NAD(P)H. In some cases, the authors are using a mixed ratio – NAD(P)H:NADP+ or similar. It isn't clear why this is important, or whether this is just an oversight? Indeed, given that the Tran et al., (2021) paper showed that NADK2 is required for P5C synthesis, can we now be more certain that ALDH18A1 is most likely to utilise NADPH under physiological conditions?

We thank the reviewer for bringing this to our attention. We have corrected this and switched to using NADH/NAD+ and NADPH/NADP+ as appropriate. We are also in agreement that NADPH is most likely to be utilized under physiological conditions for proline synthesis.

7. In Figure 4A, the BSO values appear to be close to 0, or not detected. However, as it stands this could be clearer – could the authors use another method – either split axis or log scale, or highlight 'ND' if it was below the analytical noise?

We thank the reviewer for looking for clarification on this point. Indeed, some glutathione-related metabolites are not detected (N.D.), whereas some are just at much lower levels with BSO. We have now indicated this on the graph.

8. Could the authors comment on the possibility that some of the metabolite rescue observed after eGSH treatment could be through recycling of the compound itself into metabolic pools – e.g. glutamate? (Figure S4B).

From our metabolomics data, it appears that ethyl GSH wasn't converted to endogenous GSH intracellularly. We often observe this for esterified metabolites in this cell type for an as yet unexplained reason, but it is possibly due to a lack of carboxylic acid esterases. As such, the increase in glutamate reflects more likely a decrease in the synthesis of endogenous GSH instead of a recycling of eGSH. However, we agree it would be interesting to test this directly by using labelled U-^13^C-labelled eGSH in future experiments.

9. In Figure 4G, use of oligomycin did not alter the cellular NADH:NAD+ ratio. This is a non-obvious result, as it could be expected to change, depending on the time for which oligomycin treatment was used. Could the authors comment on this in the text, or show how the cells are compensating with significantly increased lactate synthesis?

We again apologize for the oversight for not making the treatment for oligomycin more apparent and in this context an early 1 h timepoint was used. Likely, we didn't observe a change in the ratio given the rapid induction of glycolysis often observed with oligomycin and compensation via lactate synthesis. In agreement, we do observe a significant increase in the lactate to pyruvate ratio, indicating potential compensatory rewiring.

**Author response image 1. sa2fig1:** 

10. From line 266 onwards, the authors suggest that reduced ATP and NADPH availability is the likely explanation for the results observed. However, the authors did not report a change in the NADPH:NADP+ ratio (Figure 4G), so based on the evidence provided, the drop in ATP could be considered sufficient for their hypothesis.

We thank the reviewer for raising this point and we agree that the drop in mitochondrial ATP is the most likely cause of the impairment in proline synthesis in this context given the data. At the same time, we wouldn't want to rule out the possibility of reduced mitochondrial NADPH given the drop in NADPH/NADP ratio observed with TCA inhibition and recent findings (PMID: 33833463; PMID: 33888598) or direct redox linked modification of proline synthetic enzymes as mentioned in the Discussion section. Although the rescue experiments with BSO or eGSH prevented the drop in NADPH with TCA inhibition, it isn't clear if there is a compartment-specific rescue of the ratio and further work will likely be required to determine if this is the case.

11. In the proteomic analysis, the authors identify a significant change in a significant number of enzymes involved in amino acid metabolism. The change in ASNS, for example was a clearly consistent and significant change across treatments. However, the authors also specifically mention PYCR3 as being altered. While this is the case, there are others that appear more consistently upregulated but not mentioned. One example of this is PSPH/PSAT1/PHGDH/SHMT_1/2_, which are all increased in both FHi and SDHi conditions, and form a fairly coherent metabolic network that is highly compelling. As the authors will be highly aware, the provision of glycine for glutathione synthesis is likely to rely on these enzymes.

The referee raises another excellent point. The regulation of serine and glycine metabolic enzymes by ATF4 is well established and we agree that it also forms a consistent metabolic network that integrates with pathways that we have chosen to highlight and investigate in more depth. We have mentioned this aspect of the work in the revised text.

Reviewer #3 (Recommendations for the authors):Nice work.The key limitation I see is: the impact of mito defects on NEAA synthesis is well-established and clear, but this severe metabolic dependence is also a function of in vitro growth conditions. I see the point in "limitations" but it is worth explaining to the reader that NEAAs can be provided by other cell types in the context of tumor growth.Very nice data figures. I like the presentation of both signal and fractional labeling, which makes the findings/conclusions very easy to review and interpret!

We thank the referee for their comments and for their positive appraisal of our work. We have highlighted that NEAAs may be provided by other cells, such as those in the tumour stroma in the Discussion section of the manuscript. Even though it is an intrinsic limitation of working with cell cultures, it also highlights potential targetable metabolic vulnerabilities of tumour-associated stroma through modulation of diet and pharmacological agents that target certain pathways, such as proline biosynthesis.